# COMPRESSED-VFL: COMMUNICATION-EFFICIENT LEARNING WITH VERTICALLY PARTITIONED DATA

## ABSTRACT

We propose Compressed Vertical Federated Learning (C-VFL) for communication-efficient training on vertically partitioned data. In C-VFL, a server and multiple parties collaboratively train a model on their respective features utilizing several local iterations and sharing compressed intermediate results periodically. Our work provides the first theoretical analysis of the effect message compression has on distributed training over vertically partitioned data. We prove convergence of non-convex objectives to a fixed point at a rate of $O(\frac{1}{\sqrt{T}})$ when the compression error is bounded over the course of training. We provide specific requirements for convergence with common compression techniques, such as quantization and top-$k$ sparsification. Finally, we experimentally show compression can reduce communication by over $90\%$ without a significant decrease in accuracy over VFL without compression.

## 1 INTRODUCTION

Federated Learning (McMahan et al., 2017) is a distributed machine learning approach that has become of much interest in both theory (Li et al., 2020; Wang et al., 2019; Liu et al., 2020) and practice (Bonawitz et al., 2019; Rieke et al., 2020; Lim et al., 2020) in recent years. Naive distributed learning algorithms may require frequent exchanges of large amounts of data, which can lead to slow training performance (Lin et al., 2020). Further, participants may be globally distributed, with high latency network connections. To mitigate these factors, Federated Learning algorithms aim to be communication-efficient by design. Methods such as *local updates* (Moritz et al., 2016; Liu et al., 2019), where parties train local parameters for multiple iterations without communication, and message compression (Stich et al., 2018; Wen et al., 2017; Karimireddy et al., 2019) reduce message frequency and size, respectively, with little impact on training performance.

Federated Learning methods often target the case where the data among parties is distributed horizontally: each party's data shares the same features but parties hold data corresponding to different sample IDs. This is known as Horizontal Federated Learning (HFL) (Yang et al., 2019). However, there are several application areas where data is partitioned in a *vertical* manner: the parties store data on the same sample IDs but different feature spaces.

An example of a vertically partitioned setting includes a hospital, bank, and insurance company seeking to train a model to predict something of mutual interest, such as customer credit score. Each of these institutions may have data on the same individuals but store medical history, financial transactions, and vehicle accident reports, respectively. These features must remain local to the institutions due to privacy concerns, rules and regulations (e.g., GDPR, HIPAA), and/or communication network limitations. In such a scenario, Vertical Federated Learning (VFL) methods must be employed. Although VFL is less well-studied than HFL, there has been a growing interest in VFL algorithms recently (Hu et al., 2019; Gu et al., 2021; Cha et al., 2021), and VFL algorithms have important applications including risk prediction, smart manufacturing, and discovery of pharmaceuticals (Kairouz et al., 2021).

Typically in VFL, each party trains a local embedding function that maps raw data features to a meaningful vector representation, or *embedding*, for prediction tasks. For example, a neural network can be an embedding function for mapping the text of an online article to a vector space for classification (Koehrsen, 2018). Referring to Figure 1a, suppose Party 1 is a hospital with medical data

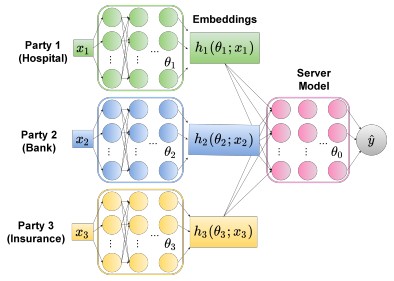

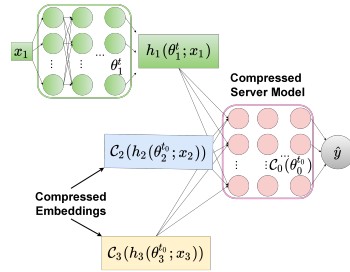

(a) Example of a global model.

(b) Local view of a global model.

Figure 1: Example global model with neural networks and its local view. **a)** To obtain a $\hat{y}$ prediction for a data sample $x$, each party $m$ feeds the local features of $x$, $x_m$, into a neural network. The output of this neural network is the embedding $h_m(\theta_m; x_m)$. All embeddings are then fed into the server model neural network with parameters $\theta_0$. **b)** When running C-VFL, Party 1 (in green) only has a compressed snapshot of the other parties embeddings and the server model. To calculate $\hat{y}$, Party 1 uses its own embedding calculated at iteration $t$, and the embeddings and server model calculated at time $t_0$, the latest communication iteration, and compressed with $\mathcal{C}_m$.

features $x_1$. The hospital computes its embedding $h_1(\theta_1; x_1)$ for the features by feeding $x_1$ through a neural network. The other parties (the bank and insurance company), compute embeddings for their features, then all parties share the embeddings in a private manner (e.g., homomorphic encryption, secure multi-party computation, or secure aggregation). The embeddings are then combined in a *server model* $\theta_0$ to determine the final loss of the global model. A server model (or fusion network) captures the complicated interactions of embeddings and is often a complex, non-linear model (Gu et al., 2019; Nie et al., 2021; Han et al., 2021). Embeddings can be very large, in practice, sometimes requiring terabytes of communication over the course of training.

Motivated by this, we propose Compressed Vertical Federated Learning (C-VFL), a general framework for communication-efficient Federated Learning over vertically partitioned data. In our algorithm, parties communicate compressed embeddings periodically, and the parties and the server each run block-coordinate descent for multiple local iterations, in parallel, using stochastic gradients to update their local parameters.

C-VFL is the first theoretically verified VFL algorithm that applies embedding compression. Unlike in HFL algorithms, C-VFL compresses embeddings rather than gradients. Previous work has proven convergence for HFL algorithms with gradient compression (Stich et al., 2018; Wen et al., 2017; Karimireddy et al., 2019). However, no previous work analyzes the convergence requirements for VFL algorithms that use embedding compression. Embeddings are parameters in the partial derivatives calculated at each party. The effect of compression error on the resulting partial derivatives may be complex; therefore, the analysis in previous work on gradient compression in HFL does not apply to compression in VFL. In our work, we prove that, under a diminishing compression error, C-VFL converges at a rate of $O(\frac{1}{\sqrt{T}})$, which is comparable to previous VFL algorithms that do not employ compression. We also analyze common compressors, such as quantization and sparsification, in C-VFL and provide bounds on their compression parameters to ensure convergence.

C-VFL also generalizes previous work by supporting an arbitrary server model. Previous work in VFL has either only analyzed an arbitrary server model without local updates (Chen et al., 2020), or analyzed local updates with a linear server model (Liu et al., 2019; Zhang et al., 2020; Das & Patterson, 2021). C-VFL is designed with an arbitrary server model, allowing support for more complex prediction tasks than those supported by previous VFL algorithms.

We summarize our main contributions in this work.

1. We introduce C-VFL with an arbitrary compression scheme. Our algorithm generalizes previous work in VFL by including both an arbitrary server model and multiple local iterations.

2. We prove convergence of C-VFL to a fixed point on non-convex objectives at a rate of $O(\frac{1}{\sqrt{T}})$ for a fixed step size when the compression error is bounded over the course of training. We also prove that the algorithm convergence error goes to zero for a diminishing step size if the compression error diminishes as well. Our work provides novel analysis for the effect of compressing embeddings on

convergence in a VFL algorithm. Our analysis also applies to Split Learning when uploads to the server are compressed.

3. We provide convergence bounds on parameters in common compressors that can be used in C-VFL. In particular, we examine scalar quantization (Bennett, 1948), lattice vector quantization (Zamir & Feder, 1996), and top-$k$ sparsification (Lin et al., 2018).

4. We evaluate our algorithm by training LSTMs on the MIMIC-III dataset and CNNs on the ModelNet10 dataset. We empirically show how C-VFL can reduce the number of bits sent by over $90\%$ compared to VFL with no compression without a significant loss in accuracy of the final model.

**Related Work.** Richtárik & Takác (2016); Hardy et al. (2017) were the first works to propose Federated Learning algorithms for vertically partitioned data. Chen et al. (2020); Romanini et al. (2021) propose the inclusion of an arbitrary server model in a VFL algorithm. However, these works do not consider multiple local iterations, and thus communicate at every iteration. Liu et al. (2019), Feng & Yu (2020), and Das & Patterson (2021) all propose different VFL algorithms with local iterations for vertically partitioned data but do not consider an arbitrary server model. In contrast to previous works, our work addresses a vertical scenario, an arbitrary server model, local iterations, and message compression.

Message compression is a common topic in HFL scenarios, where participants exchange gradients determined by their local datasets. Methods of gradient compression in HFL include scalar quantization (Bernstein et al., 2018), vector quantization (Shlezinger et al., 2021), and top-$k$ sparsification (Shi et al., 2019). In C-VFL, compressed embeddings are shared, rather than compressed gradients. Analysis in previous work on gradient compression in HFL does not apply to compression in VFL, as the effect of embedding compression error on each party's partial derivatives may be complex. No prior work has analyzed the impact of compression on convergence in VFL.

**Outline.** In Section 2, we provide the problem formulation and our assumptions. Section 3 presents the details of C-VFL. In Section 4, we present our main theoretical results. Our experimental results are given in Section 5. Finally, we conclude in Section 6.

## 2 PROBLEM FORMULATION

We present our problem formulation and notation to be used in the rest of the paper. We let $\|a\|$ be the 2-norm of a vector $a$, and let $\|\mathbf{A}\|_{\mathcal{F}}$ be the Frobenius norm of a matrix $\mathbf{A}$.

We consider a set of $M$ parties $\{1, \ldots, M\}$ and a server. The dataset $\mathbf{X} \in \mathbb{R}^{N \times D}$ is vertically partitioned a priori across the $M$ parties, where $N$ is the number of data samples and $D$ is the number of features. The $i$-th row of $\mathbf{X}$ corresponds to a data sample $x^i$. For each sample $x^i$, a party $m$ holds a disjoint subset of the features, denoted $x_m^i$, so that $x^i = [x_1^i, \ldots, x_M^i]$. For each $x^i$, there is a corresponding label $y^i$. Let $\mathbf{y} \in \mathbb{R}^{N \times 1}$ be the vector of all sample labels. We let $\mathbf{X}_m \in \mathbb{R}^{N \times D_m}$ be the local dataset of a party $m$, where the $i$-th row correspond to data features $x_m^i$. We assume that the server and all parties have a copy of the labels $\mathbf{y}$. For scenarios where the labels are private and only present at a single party, the label holder can provide enough information for the parties to compute gradients for some classes of model architectures (Liu et al., 2019).

Each party $m$ holds a set of model parameters $\theta_m$ as well as a local *embedding* function $h_m(\cdot)$. The server holds a set of parameters $\theta_0$ called the *server model* and a loss function $l(\cdot)$ that combines the *embeddings* $h_m(\theta_m; x_m^i)$ from all parties. Our objective is as follows:

$$\underset{\Theta}{\text{minimize}} \; F(\Theta; \mathbf{X}; \mathbf{y}) := \frac{1}{N} \sum_{i=1}^{N} l(\theta_0, h_1(\theta_1; x_1^i), \ldots, h_M(\theta_M; x_M^i); y^i) \qquad (1)$$

where $\Theta = [\theta_0^T, \ldots, \theta_M^T]^T$ is the *global model*. An example of a global model $\Theta$ is in Figure 1a.

For simplicity, we let $m = 0$ refer to the server, and define $h_0(\theta_0; x^i) := \theta_0$ for all $x^i$, where $h_0(\cdot)$ is equivalent to the identity function. Let $h_m(\theta_m; x_m^i) \in \mathbb{R}^{P_m}$ for $m = 0, \ldots, M$, where $P_m$ is the size of the $m$-th embedding. Let $\nabla_m F(\Theta; \mathbf{X}; \mathbf{y}) := \frac{1}{N} \sum_{i=1}^{N} \nabla_{\theta_m} l(\theta_0, h_1(\theta_1; x_1^i), \ldots, h_M(\theta_M; x_M^i); y^i)$ be the partial derivatives for parameters $\theta_m$.

Let $\mathbf{X^B}$ and $\mathbf{y^B}$ be the set of samples and labels corresponding to a randomly sampled mini-batch $\mathbf{B}$ of size $B$. We let the stochastic partial derivatives for parameters $\theta_m$ be $\nabla_m F_{\mathbf{B}}(\Theta; \mathbf{X}; \mathbf{y}) \coloneqq \frac{1}{B} \sum_{x^i, y^i \in \mathbf{X^B}, \mathbf{y^B}} \nabla_{\theta_m} l(\theta_0, h_1(\theta_1; x_1^i), \ldots, h_M(\theta_M; x_M^i); y)$. We may drop $\mathbf{X}$ and $\mathbf{y}$ from $F(\cdot)$ and $F_{\mathbf{B}}(\cdot)$. With a minor abuse of notation, we let $h_m(\theta_m; \mathbf{X}_m^{\mathbf{B}}) \coloneqq \{h_m(\theta_m; x_m^{\mathbf{B}^1}), \ldots, h_m(\theta_m; x_m^{\mathbf{B}^B})\}$ be the set of all party $m$ embeddings associated with mini-batch $\mathbf{B}$, where $\mathbf{B}^i$ is the $i$-th sample in the mini-batch $\mathbf{B}$. We let $\nabla_m F_{\mathbf{B}}(\Theta)$ and $\nabla_m F_{\mathbf{B}}(\theta_0, h_1(\theta_1; \mathbf{X}_1^{\mathbf{B}}), \ldots, h_M(\theta_M; \mathbf{X}_M^{\mathbf{B}}))$ be equivalent, and use them interchangeably.

**Assumption 1.** *Smoothness: There exists positive constants $L < \infty$ and $L_m < \infty$, for $m = 0, \ldots, M$, such that for all $\Theta_1$, $\Theta_2$, the objective function satisfies $\|\nabla F(\Theta_1) - \nabla F(\Theta_2)\| \leq L \|\Theta_1 - \Theta_2\|$ and $\|\nabla_m F_{\mathbf{B}}(\Theta_1) - \nabla_m F_{\mathbf{B}}(\Theta_2)\| \leq L_m \|\Theta_1 - \Theta_2\|$.*

**Assumption 2.** *Unbiased gradients: For $m = 0, \ldots, M$, for a randomly selected mini-batch $\mathbf{B}$, the stochastic partial derivatives are unbiased, i.e., $\mathbb{E}_{\mathbf{B}} \nabla_m F_{\mathbf{B}}(\Theta) = \nabla_m F(\Theta)$.*

**Assumption 3.** *Bounded variance: For $m = 0, \ldots, M$, there exists constants $\sigma_m < \infty$ such that the variance of the stochastic partial derivatives are bounded as: $\mathbb{E}_{\mathbf{B}} \|\nabla_m F(\Theta) - \nabla_m F_{\mathbf{B}}(\Theta)\|^2 \leq \frac{\sigma_m^2}{B}$ for a randomly selected mini-batch $\mathbf{B}$ of size $B$.*

Assumption 1 bounds how fast the gradient and stochastic partial derivatives can change. Assumptions 2 and 3 require that the stochastic partial derivatives are unbiased estimators of the true partial derivatives with bounded variance. Assumptions 1–3 are common assumptions in convergence analysis of gradient-based algorithms (Tsitsiklis et al., 1986; Nguyen et al., 2018; Bottou et al., 2018). We note Assumptions 2–3 are similar to the IID assumptions in HFL convergence analysis. However, in VFL settings, all parties store identical sample IDs but different subsets of features. Hence, there is no equivalent notion of a non-IID distribution in VFL.

**Assumption 4.** *Bounded Hessian: There exists positive constants $H_m$ for $m = 0, \ldots, M$ such that for all $\Theta$, the second partial derivatives of $F_{\mathbf{B}}$ with respect to $h_m(\theta_m; \mathbf{X}_m^{\mathbf{B}})$ satisfy: $\|\nabla^2_{h_m(\theta_m; \mathbf{X}_m^{\mathbf{B}})} F_{\mathbf{B}}(\Theta)\|_{\mathcal{F}} \leq H_m$ for any mini-batch $\mathbf{B}$.*

**Assumption 5.** *Bounded Embedding Gradients: There exists positive constants $G_m$ for $m = 0, \ldots, M$ such that for all $\theta_m$, the stochastic embedding gradients are bounded by: $\|\nabla_{\theta_m} h_m(\theta_m; \mathbf{X}_m^{\mathbf{B}})\|_{\mathcal{F}} \leq G_m$ for any mini-batch $\mathbf{B}$.*

Since we are assuming a Lipschitz-continuous loss function (Assumption 1), we know the Hessian of $F$ is bounded. Assumption 4 strengthens this assumption slightly to also bound the Hessian over any mini-batch. Assumption 5 bounds the magnitude of the partial derivatives with respect to embeddings. This embedding gradient bound is necessary to ensure convergence in the presence of embedding compression error (see appendix for details).

## 3 ALGORITHM

We now present C-VFL, a communication-efficient method for training a global model with vertically partitioned data. In each *global round*, a mini-batch $\mathbf{B}$ is chosen randomly from all samples and parties share necessary information for local training on this mini-batch. Each party, in parallel, runs block-coordinate stochastic gradient descent on its local model parameters $\theta_m$ for $Q$ local iterations. C-VFL runs for a total of $R$ global rounds, and thus runs for $T = RQ$ total local iterations.

For party $m$ to compute the stochastic gradient with respect to its features, it must receive embeddings from all parties. We reduce communication cost by only sharing embeddings every global round. Further, each party compresses their embeddings before sharing. We define a set of general compressors for compressing party embeddings and the server model: $\mathcal{C}_m(\cdot) : \mathbb{R}^{P_m} \to \mathbb{R}^{P_m}$ for $m = 0, \ldots, M$. To calculate the gradient for data sample $x^i$, party $m$ receives $\mathcal{C}_j(h_j(\theta_j; x_j^i))$ from all parties $j \neq m$. With this information, a party $m$ can compute $\nabla_m F_{\mathbf{B}}$ and update its parameters $\theta_m$ for multiple local iterations. Note that each party uses a stale view of the global model to compute its gradient during these local iterations, as it is reusing the embeddings it receives at the start of the round. In Section 4, we show that C-VFL converges even though parties use stale information. An example view a party has of the global model during training is in Figure 1b. Here, $t$ is the current iteration and $t_0$ is the start of the most recent global round, when embeddings were shared.

---

**Algorithm 1** Compressed Vertical Federated Learning

---

1: **Initialize:** $\theta_m^0$ for all parties $m$ and server model $\theta_0^0$
2: **for** $t \leftarrow 0, \ldots, T-1$ **do**
3:     **if** $t \mod Q = 0$ **then**
4:         Randomly sample $\mathbf{B}^t \in \{\mathbf{X}, \mathbf{y}\}$
5:         **for** $m \leftarrow 1, \ldots, M$ in parallel **do**
6:             Send $\mathcal{C}_m(h_m(\theta_m^t; \mathbf{X}_m^{\mathbf{B}^t}))$ to server
7:         **end for**
8:         Server sends $\{\mathcal{C}_0(\theta_0), \mathcal{C}_1(h_1(\theta_1^t; \mathbf{X}_1^{\mathbf{B}^t})), \ldots, \mathcal{C}_M(h_M(\theta_M^t; \mathbf{X}_M^{\mathbf{B}^t}))\}$ to all parties
9:     **end if**
10:     **for** $m \leftarrow 0, \ldots, M$ in parallel **do**
11:         $\hat{\Phi}_m^t \leftarrow \{\mathcal{C}_0(\theta_0^{t_0}), \mathcal{C}_1(h_1(\theta_1^{t_0}; \mathbf{X}_1^{\mathbf{B}^{t_0}})), \ldots, h_m(\theta_m^t; \mathbf{X}_m^{\mathbf{B}^{t_0}}), \ldots, \mathcal{C}_M(h_M(\theta_M^{t_0}; \mathbf{X}_M^{\mathbf{B}^{t_0}}))\}$
12:         $\theta_m^{t+1} \leftarrow \theta_m^t - \eta^{t_0} \nabla_m F_{\mathbf{B}}(\hat{\Phi}_m^t; \mathbf{y}^{\mathbf{B}^{t_0}})$
13:     **end for**
14: **end for**

---

Algorithm 1 details the procedure of C-VFL. In each global round, when $t \mod Q = 0$, a mini-batch $\mathbf{B}$ is randomly sampled from $\mathbf{X}$ and the parties exchange the associated embeddings, compressed using $\mathcal{C}_m(\cdot)$, via the server (lines 3-9). Each party $m$ completes $Q$ local iterations, using the compressed embeddings it received in iteration $t_0$ and its own $m$-th uncompressed embedding set $h_m(\theta_m^t, \mathbf{X}_m^{\mathbf{B}^{t_0}})$. We denote the set of embeddings that party $m$ uses as:

$$\hat{\Phi}_m^t \coloneqq \{\mathcal{C}_0(\theta_0^{t_0}), \mathcal{C}_1(h_1(\theta_1^{t_0}; \mathbf{X}_1^{\mathbf{B}^{t_0}})), \ldots, h_m(\theta_m^t; \mathbf{X}_m^{\mathbf{B}^{t_0}}), \ldots, \mathcal{C}_M(h_M(\theta_M^{t_0}; \mathbf{X}_M^{\mathbf{B}^{t_0}}))\}. \qquad (2)$$

For each local iteration, each party $m$ updates $\theta_m$ by computing the stochastic partial derivatives $\nabla_m F_{\mathbf{B}}(\hat{\Phi}_m^t; \mathbf{y}^{\mathbf{B}^{t_0}})$ and applying a gradient step with step size $\eta^{t_0}$ (lines 13-16).

A key difference here from previous VFL algorithms is that C-VFL shares the server model with all parties in order to support multiple local gradient updates with a non-linear server model. Also note that the same mini-batch is used for all $Q$ local iterations, thus communication is only required every $Q$ iterations. Therefore, without any compression, the total communication cost is $O(R \cdot M \cdot (B \cdot P_m + |\theta_0|))$ for $R$ global rounds. Our compression technique replaces $P_m$ and $|\theta_0|$ with smaller values based on the compression factor. For cases where embeddings, the batch size, and the server model are large, this reduction can greatly decrease the communication cost.

**Privacy.** We now discuss privacy-preserving mechanisms for C-VFL. In HFL settings, model update or gradient information is shared in messages. It has been shown that gradients can leak information about the raw data (Phong et al., 2018; Geiping et al., 2020). However in C-VFL, parties only share embeddings and can only calculate the partial derivatives associated with the server model and their local models. Commonly proposed HFL gradient attacks cannot be performed on C-VFL. Embeddings may be vulnerable to model inversion attacks (Mahendran & Vedaldi, 2015), which are methods by which an attacker can recover raw input to a model using the embedding output and black-box access to the model. One can protect against such an attack using homomorphic encryption (Cheng et al., 2019; Hardy et al., 2017) or secure multi-party computation (Gu et al., 2021). An efficient implementation of encryption for embeddings in VFL has been provided in the FATE open-source project (FederatedAI, 2021). Alternatively, if the input to the server model is the sum of party embeddings, then secure aggregation methods (Bonawitz et al., 2016) can be applied.

Note that C-VFL assumes all parties have access to the labels. For low-risk scenarios, such as predicting credit score, labels may not need to be private among the parties. In cases where labels are private, one can augment C-VFL to apply the method in Liu et al. (2019) for gradient calculation without the need for sharing labels. Our analysis in Section 4 would still hold in this case, and the additional communication is reduced by the use of message compression.

## 4 ANALYSIS

In this section, we discuss our analytical approach and present our theoretical results. We first define the compression error associated with $\mathcal{C}_m(\cdot)$:

**Definition 1.** *Compression Error: Let vectors $\epsilon_m^{x^i}$ for $m = 0, \ldots, M$, be the compression errors of $\mathcal{C}_m(\cdot)$ on a data sample $x^i$: $\epsilon_m^{x^i} := \mathcal{C}_m(h_m(\theta_m; x^i)) - h_m(\theta_m; x^i)$. Let $\epsilon_m^{t_0}$ be the $P_m \times B$ matrix with $\epsilon_m^{x^i}$ for all data samples $x^i$ in mini-batch $\boldsymbol{B}^{t_0}$ as the columns. We denote the expected squared message compression error from party $m$ at round $t_0$ as $\mathcal{E}_m^{t_0} := \mathbb{E} \|\epsilon_m^{t_0}\|_{\mathcal{F}}^2$.*

Let $\hat{\mathbf{G}}^t = [(\nabla_0 F_{\mathbf{B}}(\hat{\Phi}_0^t; \mathbf{y}^{\mathbf{B}^{t_0}}))^T, \ldots, (\nabla_M F_{\mathbf{B}}(\hat{\Phi}_M^t; \mathbf{y}^{\mathbf{B}^{t_0}}))^T]^T$. The model $\Theta$ evolves as:

$$\Theta^{t+1} = \Theta^t - \eta^{t_0} \hat{\mathbf{G}}^t. \tag{3}$$

We note the reuse of the mini-batch of $\mathbf{B}^{t_0}$ for $Q$ iterations in this recursion. This indicates that the stochastic gradients are not unbiased during local iterations $t_0 + 1 \leq t \leq t_0 + Q - 1$. However, using conditional expectation, we can apply Assumption 2 to the gradient calculated at iteration $t_0$ when there is no compression error. We define $\Phi_m^t$ to be the set of embeddings that would be received by party $m$ if no compression error were applied:

$$\Phi_m^t = \{\theta_0^{t_0}, h_1(\theta_1^{t_0}; \mathbf{X}_1^{\mathbf{B}^{t_0}}), \ldots, h_m(\theta_m^{t_0}; \mathbf{X}_m^{\mathbf{B}^{t_0}}), \ldots, h_M(\theta_M^{t_0}; \mathbf{X}_M^{\mathbf{B}^{t_0}})\}. \tag{4}$$

Then, if we take expectation over $\mathbf{B}^{t_0}$ conditioned on previous global models $\Theta^t$ up to $t_0$:

$$\mathbb{E}_{\mathbf{B}^{t_0}}[\nabla_m F_{\mathbf{B}}(\Phi_m^{t_0}) \mid \{\Theta^\tau\}_{\tau=0}^{t_0}] = \nabla_m F(\Phi_m^{t_0}). \tag{5}$$

With the help of (5), we can prove convergence by bounding the difference between the gradient at the start of each global round and those calculated during local iterations (see the proof of Lemma 2 in the appendix for details).

To account for compression error, using the chain rule and Taylor series expansion, we obtain:

**Lemma 1.** *Under Assumptions 4-5, the norm of the difference between the objective function value with compressed and uncompressed embeddings is bounded as:*

$$\mathbb{E}\|\nabla_m F_{\boldsymbol{B}}(\hat{\Phi}_m^t) - \nabla_m F_{\boldsymbol{B}}(\Phi_m^t)\|^2 \leq H_m^2 G_m^2 \sum_{j=0, j \neq m}^M \mathcal{E}_j^{t_0}. \tag{6}$$

The proof of Lemma 1 is given in the appendix. Using Lemma 1, we can bound the effect of compression error on convergence.

We present our main theoretical results. All proofs are provided in the appendix.

**Theorem 1.** *Convergence with fixed step size: Under Assumptions 1-5, if $\eta^{t_0} = \eta$ for all iterations and satisfies $\eta^{t_0} \leq \frac{1}{16Q \max\{L, \max_m L_m\}}$, then the average squared gradient over $R$ global rounds of Algorithm 1 is bounded by:*

$$\frac{1}{R} \sum_{t_0=0}^{R-1} \mathbb{E}\left[\|\nabla F(\Theta^{t_0})\|^2\right] \leq \frac{2\left[F(\Theta^0) - \mathbb{E}\left[F(\Theta^T)\right]\right]}{\eta T} + 4\eta Q L \sum_{m=0}^M \frac{\sigma_m^2}{B}$$

$$+ \frac{68Q^2}{R} \sum_{m=0}^M H_m^2 G_m^2 \sum_{t_0=0}^{R-1} \sum_{j=0, j \neq m}^M \mathcal{E}_j^{t_0}. \tag{7}$$

The first term in (7) is based on the difference between the initial model and final model of the algorithm. The second term is the error associated with the variance of the stochastic gradients and the Lipschitz constants $L$ and $L_m$'s. The third term relates to the average compression error over all iterations. The larger the error introduced by a compressor, the larger the convergence error is. We note that setting $\mathcal{E}_j^{t_0} = 0$ for all parties and iterations provides an error bound on VFL without compression and is an improvement over the bound in Liu et al. (2019) in terms of $Q$, $M$, and $B$. The second and third terms include a coefficient relating to local iterations. As the number of local iterations $Q$ increases, the convergence error increases. However, increasing $Q$ also has the effect of reducing the number of communication rounds. Thus, it may be beneficial to have $Q > 1$ in practice. We explore this more in experiments in Section 5. The second and third terms scale with $M$, the number of parties. However, VFL scenarios typically have a small number of parties (Kairouz et al., 2021), and thus $M$ plays a small role in convergence error. We note that when $M = 1$ and $Q = 1$, Theorem 1 applies to Split Learning (Gupta & Raskar, 2018) when only uploads to the server are compressed.

Table 1: Choice of common compressor parameters to achieve a convergence rate of $O(1/\sqrt{T})$. $P_m$ is the size of the $m$-th embedding. In scalar quantization, we let there be $2^q$ quantization levels, and let $h_{\max}$ and $h_{\min}$ be respectively the maximum and minimum components in $h_m(\theta_m^t; x_m^i)$ for all iterations $t$, parties $m$, and $x_m^i$. We let $V$ be the size of the lattice cell in vector quantization. We let $k$ be the number of parameters sent in an embedding after top-$k$ sparsification, and $(\|h\|^2)_{\max}$ be the maximum value of $\|h_m(\theta_m^t; x_m^i)\|^2$ for all iterations $t$, parties $m$, and $x_m^i$.

| | Uniform Scalar Quantizer | Lattice Quantization | Top-$k$ Sparsification |
|---|---|---|---|
| **Parameter choice** | $q = \Omega\left(\log_2\left(BP_m(h_{\max}-h_{\min})^2\sqrt{T}\right)\right)$ | $V = O\left(\frac{1}{BP_m\sqrt{T}}\right)$ | $k = \Omega\left(P_m - \frac{P_m}{B(\|h\|^2)_{\max}\sqrt{T}}\right)$ |
| **Compression error** | $\mathcal{E}_m^{t_0} \leq BP_m\frac{(h_{\max}-h_{\min})^2}{12}2^{-2q} = O(\frac{1}{\sqrt{T}})$ | $\mathcal{E}_m^{t_0} \leq \frac{VBP_m}{24} = O(\frac{1}{\sqrt{T}})$ | $\mathcal{E}_m^{t_0} \leq B(1-\frac{k}{P_m})(\|h\|^2)_{\max} = O(\frac{1}{\sqrt{T}})$ |

**Remark 1.** Let $\mathcal{E} = \frac{1}{R}\sum_{t_0=0}^{R-1}\sum_{m=0}^{M}\mathcal{E}_m^{t_0}$. If $\eta^{t_0} = \frac{1}{\sqrt{T}}$ for all global rounds $t_0$, for $Q$ and $B$ independent of $T$, then $\frac{1}{R}\sum_{t_0=0}^{R-1}\mathbb{E}\left[\|\nabla F(\Theta^{t_0})\|^2\right] = O(\frac{1}{\sqrt{T}}+\mathcal{E})$. This indicates that if $\mathcal{E} = O(\frac{1}{\sqrt{T}})$ then we can achieve a convergence rate of $O(\frac{1}{\sqrt{T}})$. Informally, this means that C-VFL can afford compression error and not worsen asymptotic convergence when this condition is satisfied. We discuss how this affects commonly used compressors in practice later in the section.

We consider a diminishing step size in the following theorem.

**Theorem 2.** *Convergence with diminishing step size: Under Assumptions 1-5, if $0 < \eta^{t_0} < 1$ satisfies $\eta^{t_0} \leq \frac{1}{16Q\max\{L,\max_m L_m\}}$, then the minimum squared gradient over $R$ global rounds of Algorithm 1 is bounded by:*

$$\min_{t_0=0,\ldots,R-1}\mathbb{E}\left[\|\nabla F(\Theta^{t_0})\|^2\right] = O\left(\frac{1}{\sum_{t_0=0}^{R-1}\eta^{t_0}} + \frac{\sum_{t_0=0}^{R-1}(\eta^{t_0})^2}{\sum_{t=0}^{T-1}\eta^{t_0}} + \frac{\sum_{t_0=0}^{R-1}\sum_{m=0}^{M}\eta^{t_0}\mathcal{E}_m^{t_0}}{\sum_{t_0=0}^{R-1}\eta^{t_0}}\right).$$

*If $\eta^{t_0}$ and $\mathcal{E}_m^{t_0}$ satisfy $\sum_{t_0=0}^{\infty}\eta^{t_0} = \infty$, $\sum_{t_0=0}^{\infty}(\eta^{t_0})^2 < \infty$, and $\sum_{t_0=0}^{\infty}\sum_{m=0}^{M}\eta^{t_0}\mathcal{E}_m^{t_0} < \infty$, then $\min_{t_0=0,\ldots,R-1}\mathbb{E}\left[\|\nabla F(\Theta^{t_0})\|^2\right] \to 0$ as $R \to \infty$.*

According to Theorem 2, the product of the step size and the compression error must be summable over all iterations. In the next subsection, we discuss how to choose common compressor parameters to ensure this property is satisfied. We also see in Section 5 that good results can be achieved empirically without diminishing the step size or compression error.

**Common Compressors.** In this section, we show how to choose common compressor parameters to achieve a convergence rate of $O(\frac{1}{\sqrt{T}})$ in the context of Theorem 1, and guarantee convergence in the context of Theorem 2. We analyze three common compressors: a uniform scalar quantizer (Bennett, 1948), a 2-dimensional hexagonal lattice quantizer (Zamir & Feder, 1996), and top-$k$ sparsification (Lin et al., 2018). For uniform scalar quantizer, we let there be $2^q$ quantization levels. For the lattice vector quantizer, we let $V$ be the volume of each lattice cell. For top-$k$ sparsification, we let $k$ be the number of embedding components sent in a message. In Table 1, we present the choice of compressor parameters in order to achieve a convergence rate of $O(\frac{1}{\sqrt{T}})$ in the context of Theorem 1. We show how we calculate these bounds in the appendix and provide some implementation details for their use. We can also use Table 1 to choose compressor parameters to ensure convergence in the context of Theorem 2. Let $\eta^{t_0} = O(\frac{1}{t_0})$, where $t_0$ is the current round. Then setting $T = t_0$ in Table 1 provides a choice of compression parameters at each iteration to ensure the compression error diminishes at a rate of $O(\frac{1}{\sqrt{t_0}})$, guaranteeing convergence. Diminishing compression error can be achieved by increasing the number of quantization levels, decreasing the volume of each lattice cell, or increasing the number of components sent in messages.

## 5 EXPERIMENTS

We present experiments to examine the performance of C-VFL in practice. The goal of our experiments is to examine the effects different compression techniques have on training, and investigate the accuracy/communication trade-off empirically.

Unless otherwise specified, our experimental setup consists of four parties and a server. Most real-world VFL settings include collaboration between a few institutions (Kairouz et al., 2021), so we

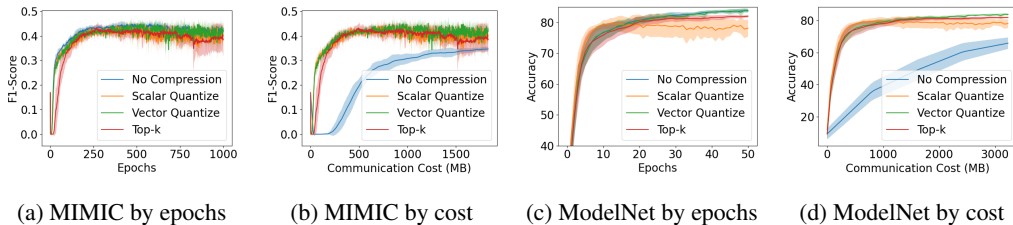

(a) MIMIC by epochs     (b) MIMIC by cost     (c) ModelNet by epochs     (d) ModelNet by cost

Figure 2: C-VFL when compressing to 2 bits per component. We show test $F_1$-Score on MIMIC-III dataset and test accuracy on ModelNet10 dataset, plotted by epochs and communication cost (MB).

expect the number of parties to be small. We train our system with two datasets: the MIMIC-III dataset (Johnson et al., 2016) and the ModelNet10 dataset (Wu et al., 2015). MIMIC-III is an anonymized hospital patient time series dataset, while ModelNet10 are CAD photos of objects, each with 12 different views. For MIMIC-III, the task is binary classification to predict in-hospital mortality. Each party trains on 19 of the 76 features with an LSTM and the server model consists of two fully-connected layers. For ModelNet10, the task is classification of images into 10 object classes. Each party trains on three views with three convolutional layers and the server model consists of a fully-connected layer. We use a fixed step size of 0.01 for the MIMIC-III dataset and 0.001 for the ModelNet10 dataset. For MIMIC-III, we use a batch size of 1000, and for ModelNet10, we use a batch size of 16. We train on the MIMIC-III dataset for 1000 epochs and the ModelNet10 dataset for 50 epochs, where an epoch consists of all iterations to fully iterate over the dataset. More details on the datasets and training procedure can be found in the appendix.

We consider the three compressors discussed in Section 4: a uniform scalar quantizer, a 2-dimensional hexagonal lattice quantizer, and top-$k$ sparsification. For both quantizers, the embedding values need to be bounded. In the case of MIMIC-III's LSTM, the embedding values are the output of a tanh activation function and have a bounded range of $[-1, 1]$. For ModelNet10, the embeddings are the output of a ReLU activation function, and may be unbounded. We scale embedding values for ModelNet10 to the range $[0, 1]$. We apply subtractive dithering to both the scalar quantizer (Wannamaker, 1997) and vector quantizer (Shlezinger et al., 2021).

In our experiments, each embedding component is a 32-bit float. Let $b$ be the bits per component we compress to. For the scalar quantizer, this means there are $2^b$ quantization levels. For the 2-D vector quantizer, this means there are $2^{2b}$ vectors in the codebook. The volume $V$ of the vector quantizer is a function of the number of codebook vectors. For top-$k$ sparsification, $k = P_m \frac{b}{32}$ as we are using 32-bit components. We train using C-VFL and consider cases where $b = 2$, $3$, and $4$. We compare with a case where $b = 32$. This corresponds to a standard VFL algorithm without embedding compression, acting as a baseline for accuracy.

In Figure 2, we plot the test $F_1$-Score and test accuracy for MIMIC-III and ModelNet10, respectively, when training with $b = 2$. We use $F_1$-Score for MIMIC-III as the in-hospital mortality prediction task is highly skewed; most people in the dataset did not die in the hospital. The solid line in each plot represents the average loss over five runs, while the shaded regions represent the standard deviation. In Figures 2a and 2c, we plot by the number of training epochs. We can see in all cases, although convergence can be a bit slower, training with compressed embeddings still reaches similar accuracy to no compression. In Figures 2b and 2d, we plot by the communication cost in MB. The cost of communication includes both the upload of (compressed) embeddings to the server and download of embeddings and server model to all parties. We can see that by compressing embeddings, we can reach higher accuracy with significantly less communication cost. In both datasets, the compressors reach similar accuracy to each other, though top-$k$ sparsification performs slightly worse than the others on MIMIC-III, while scalar quantization performs worse on ModelNet10.

In Table 2, we show the maximum test accuracy reached during training and the communication cost to reach a target accuracy for both datasets. We show results for all three compressors with $b = 2$, 3, and 4 bits per component, as well as the baseline of $b = 32$. For the MIMIC-III dataset, we show the maximum test $F_1$-score reached and the total communication cost of reaching an $F_1$-Score of 0.4. The maximum $F_1$-score for each case is within a standard deviation of each other. However, the cost to reach target score is much smaller as the value of $b$ decreases for all compressors. We can see that when $b = 2$, we can achieve over 90% communication cost reduction over no compression to reach a target $F_1$-score.

For the ModelNet10 dataset, Table 2 shows the maximum test accuracy reached and the total communication cost of reaching an accuracy of 75%. We can see similar results as for the MIMIC-III

Table 2: Maximum $F_1$-Score and test accuracy reached during training, and communication cost to reach a target accuracy. For MIMIC-III, the target test $F_1$-Score is $0.4$, For ModelNet10, the target test accuracy is $75\%$. In these experiments, $Q = 10$ and $M = 4$.

| Compressor | MIMIC-III dataset | | ModelNet10 dataset | |
|---|---|---|---|---|
| | Max $F_1$-Score Reached | Cost (MB) Target $= 0.4$ | Max Accuracy Reached | Cost (MB) Target $= 75\%$ |
| None $b = 32$ | $0.448 \pm 0.010$ | $3830.0 \pm 558.2$ | $83.81\% \pm 0.54\%$ | $9715.9 \pm 2819.3$ |
| Scalar $b = 2$ | $0.441 \pm 0.018$ | $233.1 \pm 28.7$ | $79.63\% \pm 1.74\%$ | $374.7 \pm 48.3$ |
| Vector $b = 2$ | $0.451 \pm 0.021$ | $236.1 \pm 17.9$ | $83.92\% \pm 0.66\%$ | $620.2 \pm 194.2$ |
| Top-$k$ $b = 2$ | $0.431 \pm 0.016$ | $309.8 \pm 93.6$ | $81.98\% \pm 0.43\%$ | $594.3 \pm 259.7$ |
| Scalar $b = 3$ | $0.446 \pm 0.011$ | $343.1 \pm 18.8$ | $79.47\% \pm 1.58\%$ | $581.4 \pm 106.1$ |
| Vector $b = 3$ | $0.455 \pm 0.020$ | $330.5 \pm 10.6$ | $83.85\% \pm 0.65\%$ | $930.2 \pm 264.3$ |
| Top-$k$ $b = 3$ | $0.435 \pm 0.030$ | $470.7 \pm 116.8$ | $82.64\% \pm 0.41\%$ | $833.3 \pm 355.2$ |
| Scalar $b = 4$ | $0.451 \pm 0.020$ | $456.0 \pm 87.8$ | $79.41\% \pm 2.23\%$ | $749.4 \pm 96.7$ |
| Vector $b = 4$ | $0.446 \pm 0.017$ | $446.5 \pm 21.3$ | $83.83\% \pm 0.62\%$ | $1240.3 \pm 352.4$ |
| Top-$k$ $b = 4$ | $0.453 \pm 0.014$ | $519.1 \pm 150.4$ | $83.17\% \pm 0.74\%$ | $1137.0 \pm 450.5$ |

Table 3: MIMIC-III time in seconds to reach a target $F_1$-Score for different local iterations $Q$ and communication latency $t_c$ with vector quantization and $b = 3$.

| $t_c$ | Time to Reach Target $F_1$-Score $0.45$ | | |
|---|---|---|---|
| | $Q = 1$ | $Q = 10$ | $Q = 25$ |
| 1 | $694.53 \pm 150.75$ | $470.86 \pm 235.35$ | $445.21 \pm 51.44$ |
| 10 | $1262.78 \pm 274.10$ | $512.82 \pm 256.32$ | $461.17 \pm 53.29$ |
| 50 | $3788.32 \pm 822.30$ | $699.30 \pm 349.53$ | $532.12 \pm 61.49$ |
| 200 | $13259.14 \pm 2878.04$ | $1398.60 \pm 699.05$ | $798.19 \pm 92.23$ |

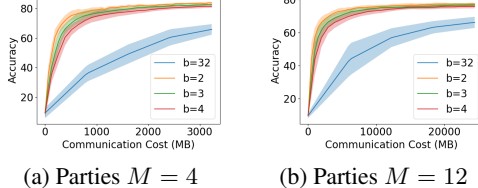

(a) Parties $M = 4$    (b) Parties $M = 12$

Figure 3: Communication cost of training on ModelNet10 with vector quantization.

dataset with the exception of scalar quantization. Scalar quantization achieves the target accuracy with much lower communication cost, but the maximum test accuracy is significantly lower than the other cases. This can be due to scalar quantization always quantizing embedding components to the same values when nearing convergence. Vector quantization benefits from considering components jointly, and thus can have better reconstruction quality than scalar quantization (Woods, 2006).

In Table 3, we consider the communication/computation tradeoff of local iterations. We show how the number of local iterations affects the time to reach a target $F_1$-score in the MIMIC-III dataset. We train C-VFL with vector quantization $b = 3$ and set the local iterations $Q$ to 1, 10, and 25. We simulate a scenario where computation time for training a mini-batch of data at each party takes 10 ms, and communication of embeddings takes a total of 1, 10, 50, and 200 ms roundtrip. These different communication latencies correspond to the distance between the parties and the server: within the same cluster, on the same local network, within the same region, and across the globe. According to Theorem 1, increasing the number of local iterations $Q$ increases convergence error. However, the target test accuracy is reached within less time when $Q$ increases. The improvement over $Q = 1$ local iterations increases as the communication latency increases. In systems where communication latency is high, it may be beneficial to increase the number of local iterations. The choice of $Q$ will depend on the accuracy requirements of the given prediction task and the time constraints on the prediction problem.

Finally, in Figure 3, we plot the test accuracy of ModelNet10 against the communication cost when using vector quantization with $b = 2, 3, 4$, and 32. We include plots for 4 and 12 parties. In the 12 party setup, each party stores one view for each CAD model. We note that changing the number of parties changes the global model structure $\Theta$ as well. We can see in both cases that smaller values of $b$ reach higher test accuracies at lower communication cost. The total communication cost is larger with 12 parties, but the impact of increasing compression is similar for both $M = 4$ and $M = 12$.

## 6 CONCLUSION

We proposed C-VFL, a distributed communication-efficient algorithm for training a model over vertically partitioned data. We proved convergence of the algorithm to a fixed point at a rate of $O(\frac{1}{\sqrt{T}})$, and we showed experimentally that communication could be reduced by over $90\%$ without a significant decrease in accuracy. For future work, we seek to relax our bounded gradient assumption and explore the effect of adaptive compressors.

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

# A APPENDIX

## A.1 PROOFS OF THEOREMS 1 AND 2

In this section, we provide the proofs for Theorems 1 and 2.

### A.1.1 ADDITIONAL NOTATION

Before starting the proofs, we define some additional notation to be used throughout. At each iteration $t$, each party $m$ trains with the embeddings $\hat{\Phi}_m^t$. This is equivalent to the party training directly with the models $\theta_m^t$ and $\theta_j^{t_0}$ for all $j \neq m$, where $t_0$ is the last communication iteration when party $m$ received the embeddings. We define:

$$\gamma_{m,j}^t = \begin{cases} \theta_j^t & m = j \\ \theta_j^{t_0} & \text{otherwise} \end{cases} \tag{A.1}$$

to represent party $m$'s view of party $j$'s model at iteration $t$. We define the column vector $\Gamma_m^t = [(\gamma_{m,0}^t)^T; \ldots; (\gamma_{m,M}^t)^T]^T$ to be party $m$'s view of the global model at iteration $t$.

We introduce some notation to help with bounding the error introduced by compression. We define $\hat{F}_{\mathbf{B}}(\Gamma_m^t)$ to be the stochastic loss with compression error for a randomly selected mini-batch $\mathbf{B}$ calculated by party $m$ at iteration $t$:

$$\hat{F}_{\mathbf{B}}(\Gamma_m^t) := F_{\mathbf{B}}\left(\theta_0^{t_0} + \epsilon_0^{t_0}, h_1(\theta_1^{t_0}; \mathbf{X}_1^{\mathbf{B}^{t_0}}) + \epsilon_1^{t_0}, \ldots, h_m(\theta_m^t; \mathbf{X}_m^{\mathbf{B}^{t_0}}), \ldots, h_M(\theta_M^{t_0}; \mathbf{X}_M^{\mathbf{B}^{t_0}}) + \epsilon_M^{t_0}\right). \tag{A.2}$$

Recall the recursion over the global model $\Theta$:

$$\Theta^{t+1} = \Theta^t - \eta^{t_0}\hat{\mathbf{G}}^t. \tag{A.3}$$

We can equivalently define $\hat{\mathbf{G}}^t$ as follows:

$$\hat{\mathbf{G}}^t = \left[(\nabla_0 \hat{F}_{\mathbf{B}}(\Gamma_0^t))^T, \ldots, (\nabla_M \hat{F}_{\mathbf{B}}(\Gamma_M^t))^T\right]^T. \tag{A.4}$$

Note that the compression error in $\hat{F}(\cdot)$ is applied to the embeddings, and not the model parameters. Thus, $F(\cdot)$ and $\hat{F}(\cdot)$ are different functions. In several parts of the proof, we need to bound the compression error in $\nabla_m \hat{F}_{\mathbf{B}}(\Gamma_m^t)$.

For our analysis, we redefine the set of embeddings for a mini-batch $\mathbf{B}$ of size $B$ from party $m$ as a matrix:

$$h_m(\theta_m; \mathbf{X}_m^{\mathbf{B}}) := \left[h_m(\theta_m; x_m^{\mathbf{B}^1}), \ldots, h_m(\theta_m; x_m^{\mathbf{B}^B})\right]. \tag{A.5}$$

$h_m(\theta_m; \mathbf{X}_m^{\mathbf{B}})$ is a matrix with dimensions $P_m \times B$ where each column is the embedding from party $m$ for a single sample in the mini-batch.

Let $P = \sum_{m=0}^M P_m$ be the sum of the sizes of all embeddings. We redefine the set of embeddings used by a party $m$ to calculate its gradient without compression error as a matrix:

$$\hat{\Phi}_m^t = \left[(\theta_0^{t_0})^T, (h_1(\theta_1^{t_0}; \mathbf{X}_1^{\mathbf{B}^{t_0}}))^T, \ldots, (h_m(\theta_m^t; \mathbf{X}_m^{\mathbf{B}^{t_0}}))^T, \ldots, (h_M(\theta_M^{t_0}; \mathbf{X}_M^{\mathbf{B}^{t_0}}))^T\right]^T. \tag{A.6}$$

$\hat{\Phi}_m^t$ is a matrix with dimensions $P \times B$ where each column is the concatenation of embeddings for all parties for a single sample in the mini-batch.

Recall the set of compression error vectors for a mini-batch $\mathbf{B}$ of size $B$ from party $m$ is the matrix:

$$\epsilon_m^{t_0} := \left[\epsilon_m^{\mathbf{B}^1}, \ldots, \epsilon_m^{\mathbf{B}^B}\right]. \tag{A.7}$$

$\epsilon_m^{t_0}$ is a matrix of dimensions $P_m \times B$ where each column is the compression error from party $m$ for a single sample in the mini-batch.

We define the compression error on each embedding used in party $m$'s gradient calculation at iteration $t$:

$$E_m^{t_0} = \left[ (\epsilon_0^{t_0})^T, \ldots, (\epsilon_{m-1}^{t_0})^T, \mathbf{0}^T, (\epsilon_{m-1}^{t_0})^T, \ldots, (\epsilon_M^{t_0})^T \right]^T. \tag{A.8}$$

$E_m^{t_0}$ is a matrix with dimensions $P \times B$ where each column is the concatenation of compression error on embeddings for all parties for a single sample in the mini-batch.

With some abuse of notation, we define:

$$\nabla_m F_{\mathbf{B}}(\Phi_m^t + E_m^{t_0}) := \nabla_m \hat{F}_{\mathbf{B}}(\Gamma_m^t). \tag{A.9}$$

Note that we can apply the chain rule to $\nabla_m \hat{F}_{\mathbf{B}}(\Gamma_m^t)$:

$$\nabla_m \hat{F}_{\mathbf{B}}(\Gamma_m^t) = \nabla_{\theta_m} h_m(\theta_m^t) \nabla_{h_m(\theta_m)} F_{\mathbf{B}}(\Phi_m^t + E_m^{t_0}). \tag{A.10}$$

With this expansion, we can now apply Taylor series expansion to $\nabla_{h_m(\theta_m)} F_{\mathbf{B}}(\Phi_m^t + E_m^{t_0})$ around the point $\Phi_m^t$:

$$\nabla_{h_m(\theta_m)} F_{\mathbf{B}}(\Phi_m^t + E_m^{t_0}) = \nabla_{h_m(\theta_m)} F_{\mathbf{B}}(\Phi_m^t) + \nabla_{h_m(\theta_m)}^2 F_{\mathbf{B}}(\Phi_m^t)^t E_m^{t_0} + \ldots \tag{A.11}$$

We let the infinite sum of all terms in this Taylor series from the second partial derivatives and up be denoted as $R_0^m$:

$$R_0^m(\Phi_m^t + E_m^{t_0}) := \nabla_{h_m(\theta_m)}^2 F_{\mathbf{B}}(\Phi_m^t)^T E_m^{t_0} + \ldots \tag{A.12}$$

Note that all compression error is in $R_0^m(\Phi_m^t + E_m^{t_0})$. Presented in Section A.1.2, the proof of Lemma 1' shows how we can bound $R_0^m(\Phi_m^t + E_m^{t_0})$, bounding the compression error in $\nabla_m \hat{F}_{\mathbf{B}}(\Gamma_m^t)$.

Let $\mathbb{E}^{t_0} = \mathbb{E}_{\mathbf{B}^{t_0}}[ \cdot \mid \{\Theta^\tau\}_{\tau=0}^{t_0}]$. Note that by Assumption 2, $\mathbb{E}^{t_0}[\mathbf{G}^{t_0}] = \nabla F(\Theta^{t_0})$ as when there is no compression error in the gradients $\mathbf{G}$, they are equal to the full-batch gradient in expectation when conditioned on the model parameters up to the iteration $t_0$. However, this is not true for iterations $t_0 + 1 \le t \le t_0 + Q - 1$, as we reuse the mini-batch $\mathbf{B}^{t_0}$ in these local iterations. We upper bound the error introduced by stochastic gradients calculated during local iterations in Lemma 2.

### A.1.2 Supporting Lemmas

Next, we provide supporting lemmas and their proofs.

We restate Lemma 1 here:

**Lemma 1.** *Under Assumptions 4-5, the norm of the difference between the objective function value with and without error is bounded by:*

$$\mathbb{E} \left\| \nabla_m F_{\boldsymbol{B}}(\hat{\Phi}_m^t) - \nabla_m F_{\boldsymbol{B}}(\Phi_m^t) \right\|^2 \le H_m^2 G_m^2 \sum_{j=0, j \ne m}^M \mathcal{E}_j^{t_0}. \tag{A.13}$$

To prove Lemma 1, we first prove the following lemma:

**Lemma 1'.** *Under Assumptions 4-5, the squared norm of the partial derivatives for party $m$'s embedding multiplied by the Taylor series terms $R_0^m(\Phi_m^t + E_m^{t_0})$ is bounded by:*

$$\left\| \nabla_{\theta_m} h_m(\theta_m^t) R_0^m(\Phi_m^t + E_m^{t_0}) \right\|^2 \le H_m^2 G_m^2 \left\| E_m^{t_0} \right\|_{\mathcal{F}}. \tag{A.14}$$

*Proof.*

$$\left\| \nabla_{\theta_m} h_m(\theta_m^t) R_0^m(\Phi_m^t + E_m^{t_0}) \right\|^2 \le \left\| \nabla_{\theta_m} h_m(\theta_m^t) \right\|_{\mathcal{F}}^2 \left\| R_0^m(\Phi_m^t + E_m^{t_0}) \right\|_{\mathcal{F}}^2 \tag{A.15}$$

$$\le H_m^2 \left\| \nabla_{\theta_m} h_m(\theta_m^t) \right\|_{\mathcal{F}}^2 \left\| E_m^{t_0} \right\|_{\mathcal{F}}^2 \tag{A.16}$$

where (A.16) follows from Assumption 4 and the following property of the Taylor series approximation error:

$$\left\| R_0^m(\Phi_m^t + E_m^{t_0}) \right\|_{\mathcal{F}} \le H_m \left\| E_m^{t_0} \right\|_{\mathcal{F}}. \tag{A.17}$$

Applying Assumption 5, we have:

$$\left\| \nabla_{\theta_m} h_m(\theta_m^t) R_0^m(\Phi_m^t + E_m^{t_0}) \right\|^2 \leq H_m^2 G_m^2 \left\| E_m^{t_0} \right\|_{\mathcal{F}}^2. \tag{A.18}$$

$\square$

We now prove Lemma 1.

*Proof.* Recall that:

$$\nabla_m \hat{F}_{\mathbf{B}}(\Gamma_m^t) = \nabla_m F_{\mathbf{B}}(\Phi_m^t + E_m^{t_0}) \tag{A.19}$$
$$= \nabla_{\theta_m} h_m(\theta_m^t) \nabla_{h_m(\theta_m)} F_{\mathbf{B}}(\Phi_m^t + E_m^{t_0}). \tag{A.20}$$

Next we apply Taylor series expansion as in (A.11):

$$\nabla_m \hat{F}_{\mathbf{B}}(\Gamma_m^t) = \nabla_{\theta_m} h_m(\theta_m^t) \left( \nabla_{h_m(\theta_m)} F_{\mathbf{B}}(\Phi_m^t) + R_0^m(\Phi_m^t + E_m^{t_0}) \right) \tag{A.21}$$
$$= \nabla_m F_{\mathbf{B}}(\Gamma_m^t) + \nabla_{\theta_m} h_m(\theta_m^t) R_0^m(\Phi_m^t + E_m^{t_0}) \tag{A.22}$$

Rearranging and applying expectation and the squared 2-norm, we can bound further:

$$\mathbb{E} \left\| \nabla_m \hat{F}_{\mathbf{B}}(\Gamma_m^t) - \nabla_m F_{\mathbf{B}}(\Gamma_m^t) \right\|^2 = \mathbb{E} \left\| \nabla_{\theta_m} h_m(\theta_m^t) R_0^m(\Phi_m^t + E_m^{t_0}) \right\|^2 \tag{A.23}$$
$$\leq H_m^2 G_m^2 \mathbb{E} \left\| E_m^{t_0} \right\|_{\mathcal{F}}^2 \tag{A.24}$$
$$= H_m^2 G_m^2 \sum_{j \neq m} \mathbb{E} \left\| \epsilon_j^{t_0} \right\|_{\mathcal{F}}^2 \tag{A.25}$$
$$= H_m^2 G_m^2 \sum_{j \neq m} \mathcal{E}_j^{t_0} \tag{A.26}$$

where (A.24) follows from Lemma 1', (A.25) follows from the definition of $E_m^{t_0}$, and (A.26) follows from Definition 1. $\square$

**Lemma 2.** *If $\eta^{t_0} \leq \frac{1}{4Q \max_m L_m}$, then under Assumptions 1-5 we can bound the conditional expected squared norm difference of gradients $\boldsymbol{G}^{t_0}$ and $\hat{\boldsymbol{G}}^t$ for iterations $t_0$ to $t_0 + Q - 1$ as follows:*

$$\sum_{t=t_0}^{t_0+Q-1} \mathbb{E}^{t_0} \left[ \left\| \hat{\boldsymbol{G}}^t - \boldsymbol{G}^{t_0} \right\|^2 \right] \leq 16Q^3(\eta^{t_0})^2 \sum_{m=0}^{M} L_m^2 \left\| \nabla_m F(\Theta^{t_0}) \right\|^2$$
$$+ 16Q^3(\eta^{t_0})^2 \sum_{m=0}^{M} L_m^2 \frac{\sigma_m^2}{B}$$
$$+ 64Q^3 \sum_{m=0}^{M} H_m^2 G_m^2 \left\| E_m^{t_0} \right\|_{\mathcal{F}}^2. \tag{A.27}$$

*Proof.*

$$\mathbb{E}^{t_0}\left[\left\|\hat{\mathbf{G}}^t - \mathbf{G}^{t_0}\right\|^2\right] = \sum_{m=0}^{M}\mathbb{E}^{t_0}\left[\left\|\nabla_m\hat{F}_{\mathbf{B}}(\Gamma_m^t) - \nabla_m F_{\mathbf{B}}(\Gamma_m^{t_0})\right\|^2\right] \tag{A.28}$$

$$= \sum_{m=0}^{M}\mathbb{E}^{t_0}\left[\left\|\nabla_m\hat{F}_{\mathbf{B}}(\Gamma_m^t) - \hat{F}_{\mathbf{B}}(\Gamma_m^{t-1}) + \nabla_m\hat{F}_{\mathbf{B}}(\Gamma_m^{t-1}) - \nabla_m F_{\mathbf{B}}(\Gamma_m^{t_0})\right\|^2\right] \tag{A.29}$$

$$\leq (1+n)\sum_{m=0}^{M}\mathbb{E}^{t_0}\left[\left\|\nabla_m\hat{F}_{\mathbf{B}}(\Gamma_m^t) - \nabla_m\hat{F}_{\mathbf{B}}(\Gamma_m^{t-1})\right\|^2\right]$$

$$+ \left(1+\frac{1}{n}\right)\sum_{m=0}^{M}\mathbb{E}^{t_0}\left[\left\|\nabla_m\hat{F}_{\mathbf{B}}(\Gamma_m^{t-1}) - \nabla_m F_{\mathbf{B}}(\Gamma_m^{t_0})\right\|^2\right] \tag{A.30}$$

$$\leq 2(1+n)\sum_{m=0}^{M}\mathbb{E}^{t_0}\left[\left\|\nabla_m F_{\mathbf{B}}(\Gamma_m^t) - \nabla_m F_{\mathbf{B}}(\Gamma_m^{t-1})\right\|^2\right]$$

$$+ 2(1+n)\sum_{m=0}^{M}\mathbb{E}^{t_0}\left[\left\|\nabla_{\theta_m}h_m(\theta_m^t)R_0^m(\Phi_m^t + E_m^{t_0}) - \nabla_{\theta_m}h_m(\theta_m^{t-1})R_0^m(\Phi_m^{t-1} + E_m^{t-1})\right\|^2\right]$$

$$+ \left(1+\frac{1}{n}\right)\sum_{m=0}^{M}\mathbb{E}^{t_0}\left[\left\|\nabla_m\hat{F}_{\mathbf{B}}(\Gamma_m^{t-1}) - \nabla_m F_{\mathbf{B}}(\Gamma_m^{t_0})\right\|^2\right] \tag{A.31}$$

$$\leq 2(1+n)\sum_{m=0}^{M}\mathbb{E}^{t_0}\left[\left\|\nabla_m F_{\mathbf{B}}(\Gamma_m^t) - \nabla_m F_{\mathbf{B}}(\Gamma_m^{t-1})\right\|^2\right]$$

$$+ 8(1+n)\sum_{m=0}^{M}H_m^2 G_m^2\left\|E_m^{t_0}\right\|^2$$

$$+ \left(1+\frac{1}{n}\right)\sum_{m=0}^{M}\mathbb{E}^{t_0}\left[\left\|\nabla_m\hat{F}_{\mathbf{B}}(\Gamma_m^{t-1}) - \nabla_m F_{\mathbf{B}}(\Gamma_m^{t_0})\right\|^2\right] \tag{A.32}$$

where (A.30) follows from the fact that $(X+Y)^2 \leq (1+n)X^2 + (1+\frac{1}{n})Y^2$ for some positive $n$ and (A.32) follows from Lemma 1'.

Applying Assumption 1 to the first term in (A.30) we have:

$$\mathbb{E}^{t_0}\left[\left\|\hat{\mathbf{G}}^t - \mathbf{G}^{t_0}\right\|^2\right] \leq 2(1+n)\sum_{m=0}^{M}L_m^2\mathbb{E}^{t_0}\left[\left\|\Gamma_m^t - \Gamma_m^{t-1}\right\|^2\right]$$

$$+ 2\left(1+\frac{1}{n}\right)\sum_{m=0}^{M}\mathbb{E}^{t_0}\left[\left\|\nabla_m\hat{F}_{\mathbf{B}}(\Gamma_m^{t-1}) - \nabla_m F_{\mathbf{B}}(\Gamma_m^{t_0})\right\|^2\right]$$

$$+ 8(1+n)\sum_{m=0}^{M}H_m^2 G_m^2\left\|E_m^{t_0}\right\|^2 \tag{A.33}$$

$$= 2(\eta^{t_0})^2(1+n)\sum_{m=0}^{M}L_m^2\mathbb{E}^{t_0}\left[\left\|\nabla_m\hat{F}_{\mathbf{B}}(\Gamma_m^{t-1})\right\|^2\right]$$

$$+ 2\left(1+\frac{1}{n}\right)\sum_{m=0}^{M}\mathbb{E}^{t_0}\left[\left\|\nabla_m\hat{F}_{\mathbf{B}}(\Gamma_m^{t-1}) - \nabla_m F_{\mathbf{B}}(\Gamma_m^{t_0})\right\|^2\right]$$

$$+ 8(1+n)\sum_{m=0}^{M}H_m^2 G_m^2\left\|E_m^{t_0}\right\|^2 \tag{A.34}$$

where (A.34) follows from the update rule $\Gamma_m^t = \Gamma_m^{t-1} - \eta^{t_0}\nabla_m\hat{F}_{\mathbf{B}}(\Gamma_m^{t-1})$.

Bounding further:

$$
\mathbb{E}^{t_0}\left[\left\|\hat{\mathbf{G}}^t - \mathbf{G}^{t_0}\right\|^2\right]
$$

$$
\leq 2(\eta^{t_0})^2\,(1+n)\sum_{m=0}^{M} L_m^2\,\mathbb{E}^{t_0}\left[\left\|\nabla_m \hat{F}_{\mathbf{B}}(\Gamma_m^{t-1}) - \nabla_m F_{\mathbf{B}}(\Gamma_m^{t_0}) + \nabla_m F_{\mathbf{B}}(\Gamma_m^{t_0})\right\|^2\right]
$$

$$
+ \left(1 + \frac{1}{n}\right)\sum_{m=0}^{M}\mathbb{E}^{t_0}\left[\left\|\nabla_m \hat{F}_{\mathbf{B}}(\Gamma_m^{t-1}) - \nabla_m F_{\mathbf{B}}(\Gamma_m^{t_0})\right\|^2\right]
$$

$$
+ 8\,(1+n)\sum_{m=0}^{M} H_m^2 G_m^2\left\|E_m^{t_0}\right\|^2 \tag{A.35}
$$

$$
\leq 4(\eta^{t_0})^2\,(1+n)\sum_{m=0}^{M} L_m^2\,\mathbb{E}^{t_0}\left[\left\|\nabla_m \hat{F}_{\mathbf{B}}(\Gamma_m^{t-1}) - \nabla_m F_{\mathbf{B}}(\Gamma_m^{t_0})\right\|^2\right]
$$

$$
+ 4(\eta^{t_0})^2\,(1+n)\sum_{m=0}^{M} L_m^2\,\mathbb{E}^{t_0}\left[\left\|\nabla_m F_{\mathbf{B}}(\Gamma_m^{t_0})\right\|^2\right]
$$

$$
+ \left(1 + \frac{1}{n}\right)\sum_{m=0}^{M}\mathbb{E}^{t_0}\left[\left\|\nabla_m \hat{F}_{\mathbf{B}}(\Gamma_m^{t-1}) - \nabla_m F_{\mathbf{B}}(\Gamma_m^{t_0})\right\|^2\right]
$$

$$
+ 8\,(1+n)\sum_{m=0}^{M} H_m^2 G_m^2\left\|E_m^{t_0}\right\|^2 \tag{A.36}
$$

$$
= \sum_{m=0}^{M}\left(4(\eta^{t_0})^2\,(1+n)\,L_m^2 + \left(1 + \frac{1}{n}\right)\right)\mathbb{E}^{t_0}\left[\left\|\nabla_m \hat{F}_{\mathbf{B}}(\Gamma_m^{t-1}) - \nabla_m F_{\mathbf{B}}(\Gamma_m^{t_0})\right\|^2\right]
$$

$$
+ 4(\eta^{t_0})^2\,(1+n)\sum_{m=0}^{M} L_m^2\,\mathbb{E}^{t_0}\left[\left\|\nabla_m F_{\mathbf{B}}(\Gamma_m^{t_0})\right\|^2\right]
$$

$$
+ 8\,(1+n)\sum_{m=0}^{M} H_m^2 G_m^2\left\|E_m^{t_0}\right\|^2. \tag{A.37}
$$

Let $n = Q$. We simplify (A.37) further:

$$
\mathbb{E}^{t_0}\left[\left\|\hat{\mathbf{G}}^t - \mathbf{G}^{t_0}\right\|^2\right]
$$

$$
\leq \sum_{m=0}^{M}\left(4(\eta^{t_0})^2\,(1+Q)\,L_m^2 + \left(1 + \frac{1}{Q}\right)\right)\mathbb{E}^{t_0}\left[\left\|\nabla_m \hat{F}_{\mathbf{B}}(\Gamma_m^{t-1}) - \nabla_m F_{\mathbf{B}}(\Gamma_m^{t_0})\right\|^2\right]
$$

$$
+ 4(\eta^{t_0})^2\,(1+Q)\sum_{m=0}^{M} L_m^2\,\mathbb{E}^{t_0}\left[\left\|\nabla_m F_{\mathbf{B}}(\Gamma_m^{t_0})\right\|^2\right]
$$

$$
+ 8\,(1+Q)\sum_{m=0}^{M} H_m^2 G_m^2\left\|E_m^{t_0}\right\|_{\mathcal{F}}^2. \tag{A.38}
$$

Let $\eta^{t_0} \leq \frac{1}{4Q \max_m L_m}$. We bound (A.38) as follows:

$$\mathbb{E}^{t_0}\left[\left\|\hat{\mathbf{G}}^t - \mathbf{G}^{t_0}\right\|^2\right] \leq \left(\frac{(1+Q)}{4Q^2} + \left(1 + \frac{1}{Q}\right)\right) \sum_{m=0}^{M} \mathbb{E}^{t_0}\left[\left\|\nabla_m \hat{F}_{\mathbf{B}}(\Gamma_m^{t-1}) - \nabla_m F_{\mathbf{B}}(\Gamma_m^{t_0})\right\|^2\right]$$

$$+ 4(\eta^{t_0})^2 (1+Q) \sum_{m=0}^{M} L_m^2 \mathbb{E}^{t_0}\left[\left\|\nabla_m F_{\mathbf{B}}(\Gamma_m^{t_0})\right\|^2\right]$$

$$+ 8(1+Q) \sum_{m=0}^{M} H_m^2 G_m^2 \left\|E_m^{t_0}\right\|_{\mathcal{F}}^2 \tag{A.39}$$

$$\leq \left(\frac{1}{2Q} + \left(1 + \frac{1}{Q}\right)\right) \sum_{m=0}^{M} \mathbb{E}^{t_0}\left[\left\|\nabla_m \hat{F}_{\mathbf{B}}(\Gamma_m^{t-1}) - \nabla_m F_{\mathbf{B}}(\Gamma_m^{t_0})\right\|^2\right]$$

$$+ 4(\eta^{t_0})^2 (1+Q) \sum_{m=0}^{M} L_m^2 \mathbb{E}^{t_0}\left[\left\|\nabla_m F_{\mathbf{B}}(\Gamma_m^{t_0})\right\|^2\right]$$

$$+ 8(1+Q) \sum_{m=0}^{M} H_m^2 G_m^2 \left\|E_m^{t_0}\right\|_{\mathcal{F}}^2 \tag{A.40}$$

$$\leq \left(1 + \frac{2}{Q}\right) \sum_{m=0}^{M} \mathbb{E}^{t_0}\left[\left\|\nabla_m \hat{F}_{\mathbf{B}}(\Gamma_m^{t-1}) - \nabla_m F_{\mathbf{B}}(\Gamma_m^{t_0})\right\|^2\right]$$

$$+ 4(\eta^{t_0})^2 (1+Q) \sum_{m=0}^{M} L_m^2 \mathbb{E}^{t_0}\left[\left\|\nabla_m F_{\mathbf{B}}(\Gamma_m^{t_0})\right\|^2\right]$$

$$+ 8(1+Q) \sum_{m=0}^{M} H_m^2 G_m^2 \left\|E_m^{t_0}\right\|_{\mathcal{F}}^2. \tag{A.41}$$

We define the following notation for simplicity:

$$A^t := \sum_{m=0}^{M} \mathbb{E}^{t_0}\left[\left\|\nabla_m \hat{F}_{\mathbf{B}}(\Gamma_m^t) - \nabla_m F_{\mathbf{B}}(\Gamma_m^{t_0})\right\|^2\right] \tag{A.42}$$

$$B_0 := 4(\eta^{t_0})^2 (1+Q) \sum_{m=0}^{M} L_m^2 \mathbb{E}^{t_0}\left[\left\|\nabla_m F_{\mathbf{B}}(\Gamma_m^{t_0})\right\|^2\right] \tag{A.43}$$

$$B_1 := 8(1+Q) \sum_{m=0}^{M} H_m^2 G_m^2 \left\|E_m^{t_0}\right\|_{\mathcal{F}}^2 \tag{A.44}$$

$$C := \left(1 + \frac{2}{Q}\right). \tag{A.45}$$

Note that we have shown that $A^t \leq CA^{t-1} + B_0 + B_1$. Therefore:

$$A^{t_0+1} \leq CA^{t_0} + (B_0 + B_1) \tag{A.46}$$

$$A^{t_0+2} \leq C^2 A^{t_0} + C(B_0 + B_1) + (B_0 + B_1) \tag{A.47}$$

$$A^{t_0+3} \leq C^3 A^{t_0} + C^2(B_0 + B_1) + C(B_0 + B_1) + (B_0 + B_1) \tag{A.48}$$

$$\vdots \tag{A.49}$$

$$A^t \leq C^{t-t_0-1} A^{t_0} + (B_0 + B_1) \sum_{k=0}^{t-t_0-2} C^k \tag{A.50}$$

$$= C^{t-t_0-1} A^{t_0} + (B_0 + B_1) \frac{C^{t-t_0-1} - 1}{C - 1}. \tag{A.51}$$

We bound the first term in (A.51) by applying Lemma 1:

$$A^{t_0} = \sum_{m=0}^{M} \mathbb{E}^{t_0} \left[ \left\| \nabla_m \hat{F}_{\mathbf{B}}(\Gamma_m^{t_0}) - \nabla_m F_{\mathbf{B}}(\Gamma_m^{t_0}) \right\|^2 \right] \tag{A.52}$$

$$\leq \sum_{m=0}^{M} H_m^2 G_m^2 \left\| E_m^{t_0} \right\|_{\mathcal{F}}^2. \tag{A.53}$$

Summing over the set of local iterations $t_0, \ldots, t_0^+$, where $t_0^+ := t_0 + Q - 1$:

$$\sum_{t=t_0}^{t_0^+} C^{t-t_0-1} A^{t_0} = A^{t_0} \sum_{t=t_0}^{t_0^+} C^{t-t_0-1} \tag{A.54}$$

$$= A^{t_0} \frac{C^Q - 1}{C - 1} \tag{A.55}$$

$$= A^{t_0} \frac{\left(1 + \frac{2}{Q}\right)^Q - 1}{\left(1 + \frac{2}{Q}\right) - 1} \tag{A.56}$$

$$= Q A^{t_0} \frac{\left(1 + \frac{2}{Q}\right)^Q - 1}{2} \tag{A.57}$$

$$\leq Q A^{t_0} \frac{e^2 - 1}{2} \tag{A.58}$$

$$\leq 4 Q A^{t_0} \tag{A.59}$$

$$\leq 4Q \sum_{m=0}^{M} H_m^2 G_m^2 \left\| E_m^{t_0} \right\|_{\mathcal{F}}^2. \tag{A.60}$$

It is left to bound the second term in (A.51) over the set of local iterations $t_0, \ldots, t_0 + Q - 1$.

$$\sum_{t=t_0}^{t_0^+} (B_0 + B_1) \frac{C^{t-t_0-1} - 1}{C - 1} \leq \sum_{t=t_0}^{t_0^+} (B_0 + B_1) \frac{C^{t-t_0-1} - 1}{C - 1} \tag{A.61}$$

$$= \frac{(B_0 + B_1)}{C - 1} \left( \sum_{t=t_0}^{t_0^+} C^{t-t_0-1} - Q \right) \tag{A.62}$$

$$= \frac{(B_0 + B_1)}{C - 1} \left( \frac{C^Q - 1}{C - 1} - Q \right) \tag{A.63}$$

$$= \frac{(B_0 + B_1)}{\left(1 + \frac{2}{Q}\right) - 1} \left( \frac{\left(1 + \frac{2}{Q}\right)^Q - 1}{\left(1 + \frac{2}{Q}\right) - 1} - Q \right) \tag{A.64}$$

$$= \frac{Q(B_0 + B_1)}{2} \left( \frac{Q\left[\left(1 + \frac{2}{Q}\right)^Q - 1\right]}{2} - Q \right) \tag{A.65}$$

$$= \frac{Q^2(B_0 + B_1)}{2} \left( \frac{\left(1 + \frac{2}{Q}\right)^Q - 1}{2} - 1 \right) \tag{A.66}$$

$$\leq \frac{Q^2(B_0 + B_1)}{2} \left( \frac{e^2 - 1}{2} - 1 \right) \tag{A.67}$$

$$\leq 2Q^2(B_0 + B_1) \tag{A.68}$$

$$\tag{A.69}$$

Plugging the values for $B_0$ and $B_1$:

$$\sum_{t=t_0}^{t_0^+} (B_0 + B_1) \frac{C^{t-t_0-1} - 1}{C - 1} \le 8Q^2(\eta^{t_0})^2 (1 + Q) \sum_{m=0}^{M} L_m^2 \mathbb{E}^{t_0} \left[ \left\| \nabla_m F_{\mathbf{B}}(\Gamma_m^{t_0}) \right\|^2 \right]$$

$$+ 16Q^2(1 + Q) \sum_{m=0}^{M} H_m^2 G_m^2 \left\| E_m^{t_0} \right\|_{\mathcal{F}}^2 \tag{A.70}$$

Applying Assumption 3 and adding in the first term in (A.51):

$$\sum_{t=t_0}^{t_0^+} A^t \le 8Q^2(\eta^{t_0})^2 (1 + Q) \sum_{m=0}^{M} L_m^2 \left\| \nabla_m F(\Theta^{t_0}) \right\|^2$$

$$+ 8Q^2(\eta^{t_0})^2 (1 + Q) \sum_{m=0}^{M} L_m^2 \frac{\sigma_m^2}{B}$$

$$+ 4(4Q^2(1 + Q) + Q) \sum_{m=0}^{M} H_m^2 G_m^2 \left\| E_m^{t_0} \right\|_{\mathcal{F}}^2 \tag{A.71}$$

$$\le 16Q^3(\eta^{t_0})^2 \sum_{m=0}^{M} L_m^2 \left\| \nabla_m F(\Theta^{t_0}) \right\|^2$$

$$+ 16Q^3(\eta^{t_0})^2 \sum_{m=0}^{M} L_m^2 \frac{\sigma_m^2}{B}$$

$$+ 64Q^3 \sum_{m=0}^{M} H_m^2 G_m^2 \left\| E_m^{t_0} \right\|_{\mathcal{F}}^2. \tag{A.72}$$

$\square$

### A.1.3   PROOF OF THEOREMS 1 AND 2

Let $t_0^+ := t_0 + Q - 1$. By Assumption 1:

$$F(\Theta^{t_0^+}) - F(\Theta^{t_0}) \le \left\langle \nabla F(\Theta^{t_0}), \Theta^{t_0^+} - \Theta^{t_0} \right\rangle + \frac{L}{2} \left\| \Theta^{t_0^+} - \Theta^{t_0} \right\|^2 \tag{A.73}$$

$$= -\left\langle \nabla F(\Theta^{t_0}), \sum_{t=t_0}^{t_0^+} \eta^{t_0} \hat{\mathbf{G}}^t \right\rangle + \frac{L}{2} \left\| \sum_{t=t_0}^{t_0^+} \eta^{t_0} \hat{\mathbf{G}}^t \right\|^2 \tag{A.74}$$

$$\le -\sum_{t=t_0}^{t_0^+} \eta^{t_0} \left\langle \nabla F(\Theta^{t_0}), \hat{\mathbf{G}}^t \right\rangle + \frac{LQ}{2} \sum_{t=t_0}^{t_0^+} (\eta^{t_0})^2 \left\| \hat{\mathbf{G}}^t \right\|^2 \tag{A.75}$$

where (A.75) follows from fact that $(\sum_{n=1}^{N} x_n)^2 \le N \sum_{n=1}^{N} x_n^2$.

We bound further:

$$
\begin{aligned}
F(\Theta^{t_0^+}) - F(\Theta^{t_0}) \leq & -\sum_{t=t_0}^{t_0^+} \eta^{t_0} \left\langle \nabla F(\Theta^{t_0}), \hat{\mathbf{G}}^t - \mathbf{G}^{t_0} \right\rangle - \sum_{t=t_0}^{t_0^+} \eta^{t_0} \left\langle \nabla F(\Theta^{t_0}), \mathbf{G}^{t_0} \right\rangle \\
& + \frac{LQ}{2} \sum_{t=t_0}^{t_0^+} (\eta^{t_0})^2 \left\| \hat{\mathbf{G}}^t - \mathbf{G}^{t_0} + \mathbf{G}^{t_0} \right\|^2 
\end{aligned}
\tag{A.76}
$$

$$
\begin{aligned}
\leq & -\sum_{t=t_0}^{t_0^+} \eta^{t_0} \left\langle \nabla F(\Theta^{t_0}), \hat{\mathbf{G}}^t - \mathbf{G}^{t_0} \right\rangle - \sum_{t=t_0}^{t_0^+} \eta^{t_0} \left\langle \nabla F(\Theta^{t_0}), \mathbf{G}^{t_0} \right\rangle \\
& + LQ \sum_{t=t_0}^{t_0^+} (\eta^{t_0})^2 \left\| \hat{\mathbf{G}}^t - \mathbf{G}^{t_0} \right\|^2 + LQ \sum_{t=t_0}^{t_0^+} (\eta^{t_0})^2 \left\| \mathbf{G}^{t_0} \right\|^2 
\end{aligned}
\tag{A.77}
$$

$$
\begin{aligned}
= & \sum_{t=t_0}^{t_0^+} \eta^{t_0} \left\langle -\nabla F(\Theta^{t_0}), \mathbf{G}^{t_0} - \hat{\mathbf{G}}^t \right\rangle - \sum_{t=t_0}^{t_0^+} \eta^{t_0} \left\langle \nabla F(\Theta^{t_0}), \mathbf{G}^{t_0} \right\rangle \\
& + LQ \sum_{t=t_0}^{t_0^+} (\eta^{t_0})^2 \left\| \hat{\mathbf{G}}^t - \mathbf{G}^{t_0} \right\|^2 + LQ \sum_{t=t_0}^{t_0^+} (\eta^{t_0})^2 \left\| \mathbf{G}^{t_0} \right\|^2 .
\end{aligned}
\tag{A.78}
$$

$$
\begin{aligned}
\leq & \frac{1}{2} \sum_{t=t_0}^{t_0^+} \eta^{t_0} \left\| \nabla F(\Theta^{t_0}) \right\|^2 \\
& + \frac{1}{2} \sum_{t=t_0}^{t_0^+} \eta^{t_0} \left\| \hat{\mathbf{G}}^t - \mathbf{G}^{t_0} \right\|^2 - \sum_{t=t_0}^{t_0^+} \eta^{t_0} \left\langle \nabla F(\Theta^{t_0}), \mathbf{G}^{t_0} \right\rangle \\
& + LQ \sum_{t=t_0}^{t_0^+} (\eta^{t_0})^2 \left\| \hat{\mathbf{G}}^t - \mathbf{G}^{t_0} \right\|^2 + LQ \sum_{t=t_0}^{t_0^+} (\eta^{t_0})^2 \left\| \mathbf{G}^{t_0} \right\|^2 
\end{aligned}
\tag{A.79}
$$

where (A.79) follows from the fact that $A \cdot B = \frac{1}{2} A^2 + \frac{1}{2} B^2 - \frac{1}{2}(A - B)^2$.

We apply the expectation $\mathbb{E}^{t_0}$ to both sides of (A.79):

$$
\begin{aligned}
\mathbb{E}^{t_0}\left[ F(\Theta^{t_0^+}) \right] - F(\Theta^{t_0}) \leq & -\frac{1}{2} \sum_{t=t_0}^{t_0^+} \eta^{t_0} \left\| \nabla F(\Theta^{t_0}) \right\|^2 + \frac{1}{2} \sum_{t=t_0}^{t_0^+} \eta^{t_0} (1 + LQ\eta^{t_0}) \mathbb{E}^{t_0}\left[ \left\| \hat{\mathbf{G}}^t - \mathbf{G}^{t_0} \right\|^2 \right] \\
& + LQ \sum_{t=t_0}^{t_0^+} (\eta^{t_0})^2 \mathbb{E}^{t_0}\left[ \left\| \mathbf{G}^{t_0} \right\|^2 \right]
\end{aligned}
\tag{A.80}
$$

$$
\begin{aligned}
\leq & -\frac{1}{2} \sum_{t=t_0}^{t_0^+} \eta^{t_0} (1 - LQ\eta^{t_0}) \left\| \nabla F(\Theta^{t_0}) \right\|^2 \\
& + \frac{1}{2} \sum_{t=t_0}^{t_0^+} \eta^{t_0} (1 + LQ\eta^{t_0}) \mathbb{E}^{t_0}\left[ \left\| \hat{\mathbf{G}}^t - \mathbf{G}^{t_0} \right\|^2 \right] + LQ \sum_{m=0}^{M} \frac{\sigma_m^2}{B} \sum_{t=t_0}^{t_0^+} (\eta^{t_0})^2
\end{aligned}
\tag{A.81}
$$

$$
\begin{aligned}
= & -\frac{Q}{2} \eta^{t_0} (1 - LQ\eta^{t_0}) \left\| \nabla F(\Theta^{t_0}) \right\|^2 \\
& + \frac{1}{2} \sum_{t=t_0}^{t_0^+} \eta^{t_0} (1 + LQ\eta^{t_0}) \mathbb{E}^{t_0}\left[ \left\| \hat{\mathbf{G}}^t - \mathbf{G}^{t_0} \right\|^2 \right] + LQ^2 (\eta^{t_0})^2 \sum_{m=0}^{M} \frac{\sigma_m^2}{B}
\end{aligned}
\tag{A.82}
$$

where (A.80) follows from applying Assumption 2 and noting that $\mathbb{E}^{t_0}\left[\mathbf{G}^{t_0}\right] = \nabla F(\Theta^{t_0})$, and (A.82) follows from Assumption 3.

Applying Lemma 2 to (A.82):

$$
\begin{aligned}
\mathbb{E}^{t_0}\left[F(\Theta^{t_0^+})\right] - F(\Theta^{t_0}) \leq &-\frac{Q}{2}\eta^{t_0}(1 - LQ\eta^{t_0})\left\|\nabla F(\Theta^{t_0})\right\|^2 \\
&+ 8Q^3(\eta^{t_0})^3(1 + LQ\eta^{t_0})\sum_{m=0}^{M} L_m^2\left\|\nabla_m F(\Theta_m^{t_0})\right\|^2 \\
&+ 8Q^3(\eta^{t_0})^3(1 + LQ\eta^{t_0})\sum_{m=0}^{M} L_m^2\frac{\sigma_m^2}{B} \\
&+ 32Q^3\eta^{t_0}(1 + LQ\eta^{t_0})\sum_{m=0}^{M} H_m^2 G_m^2\left\|E_m^{t_0}\right\|_{\mathcal{F}}^2 \\
&+ LQ^2(\eta^{t_0})^2\sum_{m=0}^{M}\frac{\sigma_m^2}{B} \quad\quad\quad\quad (A.83)
\end{aligned}
$$

$$
\begin{aligned}
\leq &-\frac{Q}{2}\sum_{m=0}^{M}\eta^{t_0}(1 - LQ\eta^{t_0} - 16Q^2L_m^2(\eta^{t_0})^2 - 16Q^3L_m^2 L(\eta^{t_0})^3))\left\|\nabla_m F(\Theta^{t_0})\right\|^2 \\
&+ (LQ^2(\eta^{t_0})^2 + 8Q^3L_m^2(\eta^{t_0})^3 + 8Q^4LL_m^2(\eta^{t_0})^4)\sum_{m=0}^{M}\frac{\sigma_m^2}{B} \\
&+ 32Q^3\eta^{t_0}(1 + LQ\eta^{t_0})\sum_{m=0}^{M} H_m^2 G_m^2\left\|E_m^{t_0}\right\|_{\mathcal{F}}^2 . \quad\quad\quad\quad (A.84)
\end{aligned}
$$

Let $\eta^{t_0} \leq \frac{1}{16Q\max\{L,\max_m L_m\}}$. Then we bound (A.84) further:

$$
\begin{aligned}
\mathbb{E}^{t_0}\left[F(\Theta^{t_0^+})\right] - F(\Theta^{t_0}) \leq &-\frac{Q}{2}\sum_{m=0}^{M}\eta^{t_0}(1 - \frac{1}{16} - \frac{1}{16} - \frac{1}{16^2}))\left\|\nabla_m F(\Theta^{t_0})\right\|^2 \\
&+ (LQ^2(\eta^{t_0})^2 + 8Q^3L_m^2(\eta^{t_0})^3 + 8Q^4LL_m^2(\eta^{t_0})^4)\sum_{m=0}^{M}\frac{\sigma_m^2}{B} \\
&+ 16Q^3\eta^{t_0}(1 + LQ\eta^{t_0})\sum_{m=0}^{M} H_m^2 G_m^2\left\|E_m^{t_0}\right\|_{\mathcal{F}}^2 \quad\quad\quad\quad (A.85) \\
\leq &-\frac{Q}{2}\eta^{t_0}\left\|\nabla F(\Theta^{t_0})\right\|^2 \\
&+ (LQ^2(\eta^{t_0})^2 + 8Q^3L_m^2(\eta^{t_0})^3 + 8Q^4LL_m^2(\eta^{t_0})^4)\sum_{m=0}^{M}\frac{\sigma_m^2}{B} \\
&+ 32Q^3\eta^{t_0}(1 + LQ\eta^{t_0})\sum_{m=0}^{M} H_m^2 G_m^2\left\|E_m^{t_0}\right\|_{\mathcal{F}}^2 \quad\quad\quad\quad (A.86)
\end{aligned}
$$

After some rearranging of terms:

$$\eta^{t_0} \left\| \nabla F(\Theta^{t_0}) \right\|^2 \leq \frac{2 \left[ F(\Theta^{t_0}) - \mathbb{E}^{t_0} \left[ F(\Theta^{t_0^+}) \right] \right]}{Q}$$

$$+ 2(LQ(\eta^{t_0})^2 + 8Q^2 L_m^2 (\eta^{t_0})^3 + 8Q^3 L L_m^2 (\eta^{t_0})^4) \sum_{m=0}^{M} \frac{\sigma_m^2}{B}$$

$$+ 64Q^2 \eta^{t_0} (1 + LQ\eta^{t_0}) \sum_{m=0}^{M} H_m^2 G_m^2 \left\| E_m^{t_0} \right\|_{\mathcal{F}}^2 \qquad (A.87)$$

Summing over all communication rounds $t_0 = 0, \ldots, R-1$ and taking total expectation:

$$\sum_{t_0=0}^{R-1} \eta^{t_0} \mathbb{E} \left[ \left\| \nabla F(\Theta^{t_0}) \right\|^2 \right] \leq \frac{2 \left[ F(\Theta^0) - \mathbb{E} \left[ F(\Theta^T) \right] \right]}{Q}$$

$$+ 2 \sum_{t_0=0}^{R-1} (LQ(\eta^{t_0})^2 + 8Q^2 L_m^2 (\eta^{t_0})^3 + 8Q^3 L L_m^2 (\eta^{t_0})^4) \sum_{m=0}^{M} \frac{\sigma_m^2}{B}$$

$$+ 64Q^2 \eta^{t_0} (1 + LQ\eta^{t_0}) \sum_{m=0}^{M} H_m^2 G_m^2 \left\| E_m^{t_0} \right\|_{\mathcal{F}}^2 \qquad (A.88)$$

$$\leq \frac{2 \left[ F(\Theta^0) - \mathbb{E} \left[ F(\Theta^T) \right] \right]}{QR}$$

$$+ 2 \sum_{t_0=0}^{R-1} (QL(\eta^{t_0})^2 + 8Q^2 L_m^2 (\eta^{t_0})^3 + 8Q^3 L L_m^2 (\eta^{t_0})^4) \sum_{m=0}^{M} \frac{\sigma_m^2}{B}$$

$$+ 64Q^2 \sum_{t_0=0}^{R-1} \eta^{t_0} (1 + LQ\eta^{t_0}) \sum_{m=0}^{M} H_m^2 G_m^2 \mathbb{E} \left[ \left\| E_m^{t_0} \right\|_{\mathcal{F}}^2 \right]. \qquad (A.89)$$

Note that:

$$\sum_{m=0}^{M} H_m^2 G_m^2 \mathbb{E} \left[ \left\| E_m^{t_0} \right\|_{\mathcal{F}}^2 \right] = \sum_{m=0}^{M} H_m^2 G_m^2 \sum_{j \neq m} \mathbb{E} \left[ \left\| \epsilon_j^{t_0} \right\|_{\mathcal{F}}^2 \right] \qquad (A.90)$$

$$= \sum_{m=0}^{M} H_m^2 G_m^2 \sum_{j \neq m} \mathcal{E}_j^{t_0} \qquad (A.91)$$

where (A.91) follows from Definition 1.

Plugging this into (A.89)

$$\sum_{t_0=0}^{R-1} \eta^{t_0} \mathbb{E} \left[ \left\| \nabla F(\Theta^{t_0}) \right\|^2 \right] \leq \frac{2 \left[ F(\Theta^0) - \mathbb{E} \left[ F(\Theta^T) \right] \right]}{QR}$$

$$+ 2 \sum_{t_0=0}^{R-1} (QL(\eta^{t_0})^2 + 8Q^2 L_m^2 (\eta^{t_0})^3 + 8Q^3 L L_m^2 (\eta^{t_0})^4) \sum_{m=0}^{M} \frac{\sigma_m^2}{B}$$

$$+ 64Q^2 \sum_{t_0=0}^{R-1} \eta^{t_0} (1 + LQ\eta^{t_0}) \sum_{m=0}^{M} H_m^2 G_m^2 \sum_{j \neq m} \mathcal{E}_j^{t_0}. \qquad (A.92)$$

Suppose that $\eta^{t_0} = \eta$ for all communication rounds $t_0$. Then, averaging over $R$ communication rounds, we have:

$$\frac{1}{R}\sum_{t_0=0}^{R-1}\mathbb{E}\left[\left\|\nabla F(\Theta^{t_0})\right\|^2\right] \leq \frac{2\left[F(\Theta^0) - \mathbb{E}\left[F(\Theta^T)\right]\right]}{QR\eta} + 2\sum_{m=0}^{M}(QL\eta + 8Q^2L_m^2\eta^2 + 8Q^3LL_m^2\eta^3)\frac{\sigma_m^2}{B}$$

$$+ \frac{64Q^2}{R}\sum_{t_0=0}^{R-1}(1 + LQ\eta)\sum_{m=0}^{M}H_m^2G_m^2\sum_{j\neq m}\mathcal{E}_j^{t_0}. \tag{A.93}$$

$$\leq \frac{2\left[F(\Theta^0) - \mathbb{E}\left[F(\Theta^T)\right]\right]}{QR\eta} + 4\sum_{m=0}^{M}QL\eta\frac{\sigma_m^2}{B}$$

$$+ \frac{68Q^2}{R}\sum_{t_0=0}^{R-1}\sum_{m=0}^{M}H_m^2G_m^2\sum_{j\neq m}\mathcal{E}_j^{t_0}. \tag{A.94}$$

where (A.94) follows from our assumption that $\eta^{t_0} \leq \frac{1}{16Q\max\{L,\max_m L_m\}}$. This completes the proof of Theorem 1.

We continue our analysis to prove Theorem 2. Starting from (A.92), we bound the left-hand side with the minimum over all iterations:

$$\min_{t_0=0,\dots,R-1}\mathbb{E}\left[\left\|\nabla F(\Theta^{t_0})\right\|^2\right] \leq \frac{2\left[F(\Theta^0) - \mathbb{E}^{t_0}\left[F(\Theta^T)\right]\right]}{Q\sum_{t_0=0}^{R-1}\eta^{t_0}}$$

$$+ 2\left(QL\frac{\sum_{t_0=0}^{R-1}(\eta^{t_0})^2}{\sum_{t_0=0}^{R-1}\eta^{t_0}} + 8Q^2L_m^2\frac{\sum_{t_0=0}^{R-1}(\eta^{t_0})^3}{\sum_{t_0=0}^{R-1}\eta^{t_0}} + 8Q^3LL_m^2\frac{\sum_{t_0=0}^{R-1}(\eta^{t_0})^4}{\sum_{t_0=0}^{R-1}\eta^{t_0}}\right)\sum_{m=0}^{M}\frac{\sigma_m^2}{B}$$

$$+ 64Q^2\sum_{m=0}^{M}H_m^2G_m^2\frac{\sum_{t_0=0}^{R-1}\eta^{t_0}\sum_{j\neq m}\mathcal{E}_j^{t_0}}{\sum_{t_0=0}^{R-1}\eta^{t_0}} + 64LQ^3\sum_{m=0}^{M}H_m^2G_m^2\frac{\sum_{t_0=0}^{R-1}(\eta^{t_0})^2\sum_{j\neq m}\mathcal{E}_j^{t_0}}{\sum_{t_0=0}^{R-1}\eta^{t_0}}$$

$$\tag{A.95}$$

As $R \to \infty$, if $\sum_{t_0=0}^{R-1}\eta^{t_0} = \infty$, $\sum_{t_0=0}^{R-1}(\eta^{t_0})^2 < \infty$, and $\sum_{t_0=0}^{R-1}\eta^{t_0}\sum_{j\neq m}\mathcal{E}_j^{t_0} < \infty$, then $\min_{t_0=0,\dots,R-1}\mathbb{E}\left[\left\|\nabla F(\Theta^{t_0})\right\|^2\right] \to 0$. This completes the proof of Theorem 2.

## A.2 COMMON COMPRESSORS

In this section, we calculate the compression error and parameter bounds for uniform scalar quantization, lattice vector quantization and top-$k$ sparsification, as well as discuss implementation details of these compressors in C-VFL.

We first consider a uniform scalar quantizer (Bennett, 1948) with a set of $2^q$ quantization levels, where $q$ is the number of bits to represent compressed values. We define the range of values that quantize to the same quantization level as the quantization bin. In C-VFL, a scalar quantizer quantizes each individual component of embeddings. The error in each embedding of a batch $\mathbf{B}$ in scalar quantization is $\leq P_m\frac{\Delta^2}{12} = P_m\frac{(h_{max}-h_{min})^2}{12}2^{-2q}$ where $\Delta$ the size of a quantization bin, $P_m$ is the size of the $m$-th embedding, $h_{max}$ and $h_{min}$ are respectively the maximum and minimum value $h_m(\theta_m^t; x_m^i)$ can be for all iterations $t$, parties $m$, and $x_m^i$. We note that if $h_{max}$ or $h_{min}$ are unbounded, then the error is unbounded as well. By Theorem 1, we know that $\frac{1}{R}\sum_{t_0=0}^{R-1}\sum_{m=0}^{M}\mathcal{E}_m^{t_0} = O(\frac{1}{\sqrt{T}})$ to obtain a convergence rate of $O(\frac{1}{\sqrt{T}})$. If we use the same $q$ for all parties and iterations, we can solve for $q$ to find that the value $q$ must be lower bounded by $q = \Omega(\log_2(P_m(h_{max} - h_{min})^2\sqrt{T}))$ to reach a convergence rate of $O(\frac{1}{\sqrt{T}})$. For a diminishing compression error, required by Theorem 2, we let $T = t_0$ in this bound, indicating that $q$, the number of quantization bins, must increase as training continues.

A vector quantizer creates a set of $d$-dimensional vectors called a codebook (Zamir & Feder, 1996). A vector is quantized by dividing the components into sub-vectors of size $d$, then quantizing each sub-vector to the nearest codebook vector in Euclidean distance. A cell in vector quantization is

defined as all points in $d$-space that quantizes to a single codeword. The volume of these cells are determined by how closely packed codewords are. We consider the commonly applied 2-dimensional hexagonal lattice quantizer (Shlezinger et al., 2021). In C-VFL, each embedding is divided into sub-vectors of size two, scaled to the unit square, then quantized to the nearest vector by Euclidean distance in the codebook. The error in this vector quantizer is $\leq \frac{VP_m}{24}$ where $V$ is the volume of a lattice cell. The more bits available for quantization, the smaller the volume of the cells, the smaller the compression error. We can calculate an upper bound on $V$ based on Theorem 1: $V = O(\frac{1}{P_m\sqrt{T}})$. If a diminishing compression error is required, we can set $T = t_0$ in this bound, indicating that $V$ must decrease at a rate of $O(\frac{1}{P_m\sqrt{t_0}})$. As the number of iterations increases, the smaller $V$ must be, and thus the more bits that must be communicated.

In top-$k$ sparsification (Lin et al., 2018), when used in distributed SGD algorithms, the $k$ largest magnitude components of the gradient are sent while the rest are set to zero. In the case of embeddings in C-VFL, a large element may be as important as an input to the server model as a small one. We can instead select the $k$ embedding elements to send with the largest magnitude partial derivatives in $\nabla_{\theta_m} h_m(\theta_m^t)$. Since a party $m$ cannot calculate $\nabla_{\theta_m} h_m(\theta_m^t)$ until all parties send their embeddings, party $m$ can use the embedding gradient calculated in the previous iteration, $\nabla_{\theta_m} h_m(\theta_m^{t-1})$. This is an intuitive method, as we assume our gradients are Lipschitz continuous, and thus do not change too rapidly. The error of sparsification is $\leq (1 - \frac{k}{P_m})(\|h\|^2)_{max}$ where $(\|h\|^2)_{max}$ is the maximum value of $\|h_m(\theta_m^t; x_m^i)\|^2$ for all iterations $t$, parties $m$, and $x_m^i$. Note that if $(\|h\|^2)_{max}$ is unbounded, then the error is unbounded. We can calculate a lower bound on $k$: $k = \Omega(P_m - \frac{P_m}{(\|h\|^2)_{max}\sqrt{T}})$. Note that the larger $(\|h\|^2)_{max}$, the larger $k$ must be. More components must be sent if embedding magnitude is large in order to achieve a convergence rate of $O(\frac{1}{\sqrt{T}})$. When considering a diminishing compression error, we set $T = t_0$, showing that $k$ must increase over the course of training.

## A.3 Experimental Details

For our experiments, we used an internal cluster of 40 compute nodes running CentOS 7 each with $2\times$ 20-core 2.5 GHz Intel Xeon Gold 6248 CPUs, $8\times$ NVIDIA Tesla V100 GPUs with 32 GB HBM, and 768 GB of RAM.

### A.3.1 MIMIC-III

The MIMIC-III dataset can be found at: mimic.physionet.org. The dataset consists of time-series data from ~60,000 intensive care unit admissions. The data includes many features about each patient, such as demographic, vital signs, medications, and more. All the data is anonymized. In order to gain access to the dataset, one must take the short online course provided on their website.

Our code for training with the MIMIC-III dataset can be found in in the folder titled "mimic3". This is an extension of the MIMIC-III benchmarks repo found at: github.com/YerevaNN/mimic3-benchmarks. The original code preprocesses the MIMIC-III dataset and provides starter code for training LSTMs using centralized SGD. Our code has updated their existing code to TensorFlow 2. The new file of interest in our code base is "mimic3models/in_hospital_mortality/quant.py" which runs C-VFL. Both our code base and the original are under the MIT License. More details on installation, dependencies, and running our experiments can be found in "README.md". Each experiment took approximately six hours to run on a node in our cluster.

The benchmarking preprocessing code splits the data up into different prediction cases. Our experiments train models to predict for in-hospital mortality. For in-hospital mortality, there are 14,681 training samples, and 3,236 test samples. In our experiments, we use a step size of 0.01, as is standard for training an LSTM on the MIMIC-III dataset.

### A.3.2 ModelNet10

Details on the ModelNet10 dataset can be found at: modelnet.cs.princeton.edu/. The specific link we downloaded the dataset from is the following Google Drive link: https://drive.google.com/file/d/0B4v2jR3WsindMUE3N2xiLVpyLW8/view. The dataset consists of 3D CAD models of different common objects in the world. For each CAD model, there are 12 views

from different angles saved as PNG files. We only trained our models on the following 10 classes: bathtub, bed, chair, desk, dresser, monitor, night_stand, sofa, table, toilet. We used a subset of the data with $1,008$ training samples and $918$ test samples. In our experiments, we use a step size of $0.001$, as is standard for training a CNN on the ModelNet10 dataset.

Our code for learning on the ModelNet10 dataset is in the folder "MVCNN_Pytorch" and is an extension of the MVCNN-PyTorch repo: github.com/RBirkeland/MVCNN-PyTorch. The file of interest in our code base is "quant.py" which runs C-VFL. Both our code base and the original are under the MIT License. Details on how to run our experiments can be found in the "README.md". Each experiment took approximately six hours to run on a node in our cluster.

### A.4    ADDITIONAL EXPERIMENTS

In this section we provide some additional experiments to test the scalability of C-VFL for a larger number of parties and larger datasets.

First, we run C-VFL using the same parameters as described in Section 5, now with $48$ parties and a server with the ModelNet10 dataset. Each 3D CAD model in the ModelNet10 dataset has $12$ views, so we assign every four parties the same view, and have each store a different quadrant of the image.

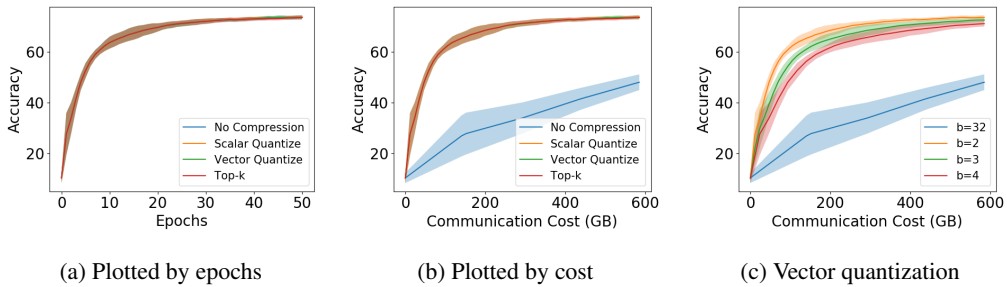

| (a) Plotted by epochs | (b) Plotted by cost | (c) Vector quantization |

Figure A.1: Test accuracy on ModelNet10 dataset with the number of parties $M = 48$. In the first two plots, the compressors have $b = 2$, where $b$ is the number of bits used to represent embedding components. In the third plot, $b = 32$ indicates there is no compression. The results show little variation between compressors and no compression, leading to a large benefit in communication cost versus test accuracy.

In Figure A.1, we plot the test accuracy for the ModelNet10 dataset. The test accuracy is overall lower than when running with $4$ and $12$ parties. This is expected, as each party has less information individually, making the prediction task more difficult. Figure A.1a shows the test accuracy plotted by epochs. There is very little variation between compressors and no compression here. This leads to a very large benefit for compression when plotting by communication cost, seen in Figure A.1b. In Figure A.1c, we plot the test accuracy of C-VFL using vector quantization for different values of $b$, the number of bits to represent compressed values. Similar to previous results, lower $b$ tends to improve test accuracy reached with the same amount of communication cost. We can also see that the total cost of communication has increased compared to the case of $4$ and $12$ parties in Figure 3. This is expected, as there are more embeddings being exchanged in each global round.

We also run C-VFL on CIFAR-10, a large dataset of $60,000$ images with $10$ classes of objects and animals. To simulate a VFL scenario with the CIFAR-10 dataset, we split the images into $4$ quadrants and run C-VFL with $4$ parties and a server. Each party trains ResNet-18 locally, and the server model is a single fully-connected layer.

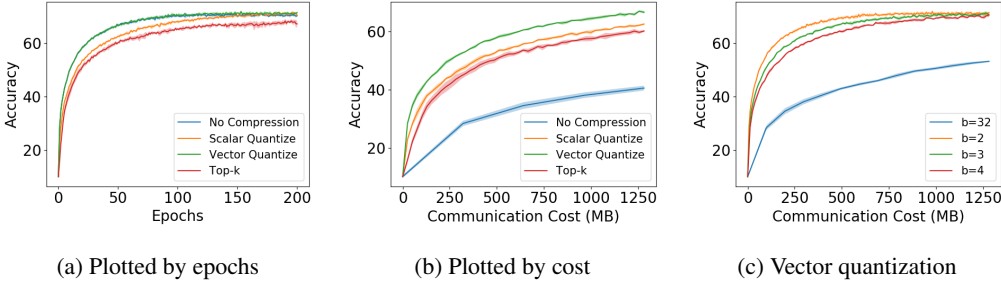

(a) Plotted by epochs        (b) Plotted by cost        (c) Vector quantization

Figure A.2: Test accuracy on CIFAR-10 dataset with the number of parties $M = 4$. In the first two plots, the compressors have $b = 2$, where $b$ is the number of bits used to represent embedding components. In the third plot, $b = 32$ indicates there is no compression. The results show vector quantization performs the best our of the compressors, and all compressors show improvement over no compression in terms of communication cost to reach target test accuracies.

In Figure A.2, we plot the test accuracy for the CIFAR-10 dataset. The test accuracy is fairly low compared to typical baseline accuracies, which is expected, as learning object classification from only a quadrant of a $32 \times 32$ pixel image is difficult. Figure A.2a shows the test accuracy plotted by epochs. We can see that vector quantization performs almost as well as no compression in the CIFAR-10 dataset. When plotting by communication cost, seen in Figure A.2b, we can see that vector quantization performs the best, though scalar quantization and top-$k$ sparsification show communication savings as well. In Figure A.2c, we plot the test accuracy of C-VFL using vector quantization for different values of $b$, the number of bits to represent compressed values. Similar to previous results, lower $b$ tends to improve test accuracy reached with the same amount of communication cost.

## A.5 Additional Plots

In this section, we include additional plots using the results from the experiments introduced in Section 5 of the main paper. The setup for the experiments is described in the main paper. These plots provide some additional insight into the effect of each compressor on convergence in both datasets. As with the plots in the main paper, the solid lines in each plot are the average of five runs and the shaded regions represent the standard deviation.

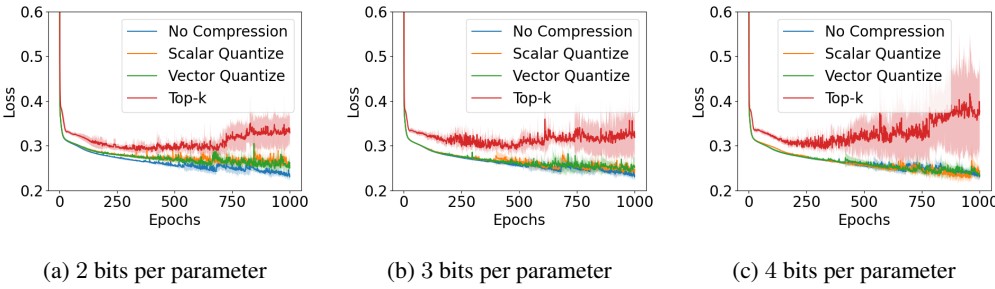

(a) 2 bits per parameter      (b) 3 bits per parameter      (c) 4 bits per parameter

Figure A.3: Training loss on MIMIC-III dataset. One can see that, with the exception of top-$k$ sparsification, allowing more bits for compression moves the training loss closer to the baseline of no compression. Top-$k$ appears to be more unstable in the MIMIC-III dataset.

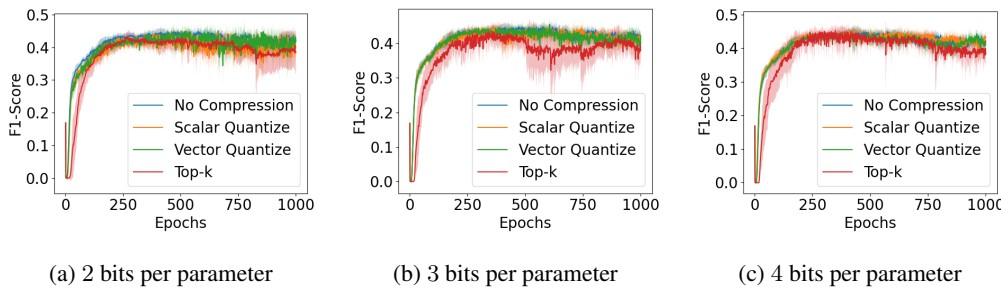

(a) 2 bits per parameter      (b) 3 bits per parameter      (c) 4 bits per parameter

Figure A.4: Test $F_1$-Score on MIMIC-III dataset. Scalar and vector quantization achieve similar test $F_1$-score even when only using 2 bits in quantization. On the other hand, top-$k$ sparsification performs worse than the other compressors in the MIMIC-III dataset.

Figures A.3 and A.4 plot the training loss and test $F_1$-Score for training on the MIMIC-III dataset for different levels of compression. We can see that scalar and vector quantization perform similarly to no compression and improve as the number of bits available increase. We can also see that top-$k$ sparsification has high variability on the MIMIC-III dataset and generally performs worse than the other compressors.

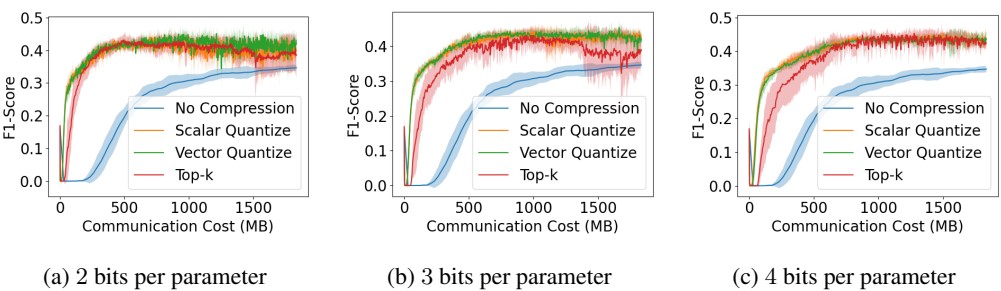

(a) 2 bits per parameter      (b) 3 bits per parameter      (c) 4 bits per parameter

Figure A.5: Test $F_1$-Score on MIMIC-III dataset plotted by communication cost. We can see that all compressors reach higher $F_1$-scores with lower communication cost than no compression. We can see that the standard deviation for each compressor decreases as the number of bits available increases. Top-$k$ sparsification generally performs worse than the other compressors on the MIMIC-III-dataset.

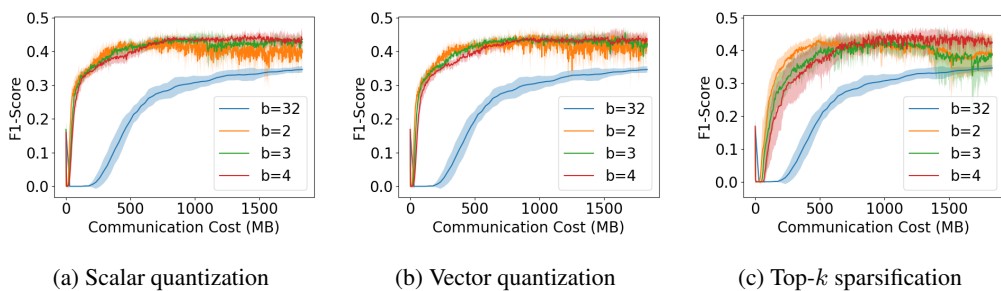

(a) Scalar quantization      (b) Vector quantization      (c) Top-$k$ sparsification

Figure A.6: Test $F_1$-Score on MIMIC-III dataset plotted by communication cost. We can see that all compressors reach higher $F_1$-scores with lower communication cost than no compression. We can see that the standard deviation for each compressor decreases as the number of bits available increases. Top-$k$ sparsification generally performs worse than the other compressors on the MIMIC-III-dataset.

Figures A.5 and A.6 plot the test $F_1$-Score for training on the MIMIC-III dataset plotted against the communication cost. The plots in Figure A.5 include all compression techniques for a given level of

compression, while the plots in Figure A.6 include all levels of compression for a given compression technique. We can see that all compressors reach higher $F_1$-scores with lower communication cost than no compression. It is interesting to note that increasing the number of bits per parameter reduces the variability in all compressors.

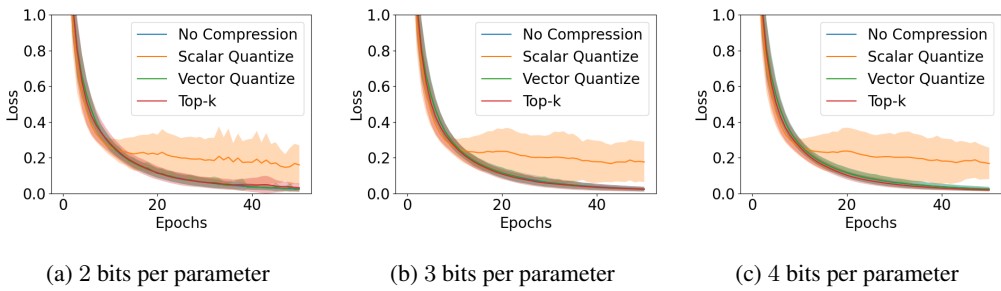

(a) 2 bits per parameter    (b) 3 bits per parameter    (c) 4 bits per parameter

Figure A.7: Training loss on ModelNet10 dataset. Vector quantization and top-$k$ sparsification perform similarly to no compression, even when only 2 bits are available. Scalar quantization converges to a higher loss and has high variability on the ModelNet10 dataset.

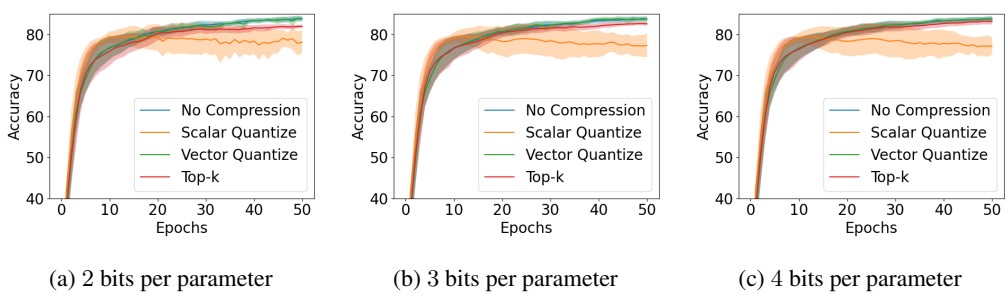

(a) 2 bits per parameter    (b) 3 bits per parameter    (c) 4 bits per parameter

Figure A.8: Test accuracy on ModelNet10 dataset. Vector quantization and top-$k$ sparsification perform similarly to no compression, even when only 2 bits are available. Scalar quantization converges to a lower test accuracy, and has high variability on the ModelNet10 dataset.

Figures A.7 and A.8 plot the training loss and test accuracy for training on the ModelNet10 dataset. Vector quantization and top-$k$ sparsification perform similarly to no compression in both training loss and test accuracy, even when only 2 bits are available. We can see that scalar quantization has high variability on the ModelNet10 dataset.

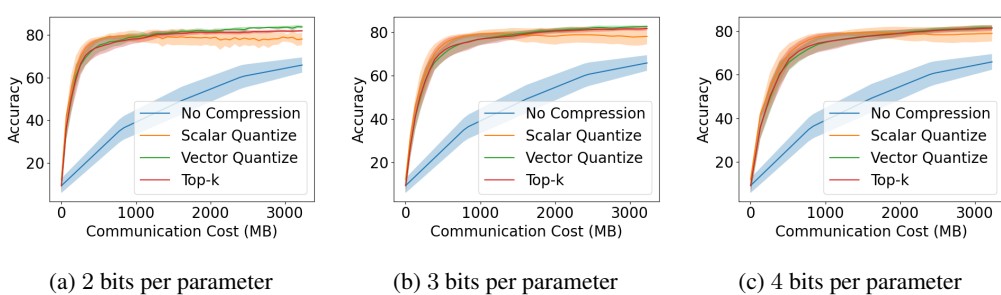

(a) 2 bits per parameter    (b) 3 bits per parameter    (c) 4 bits per parameter

Figure A.9: Test accuracy on ModelNet10 dataset plotted by communication cost. We can see that all compressors reach higher accuracies with lower communication cost than no compression. Scalar quantization generally performs worse than the other compressors on the ModelNet10 dataset.

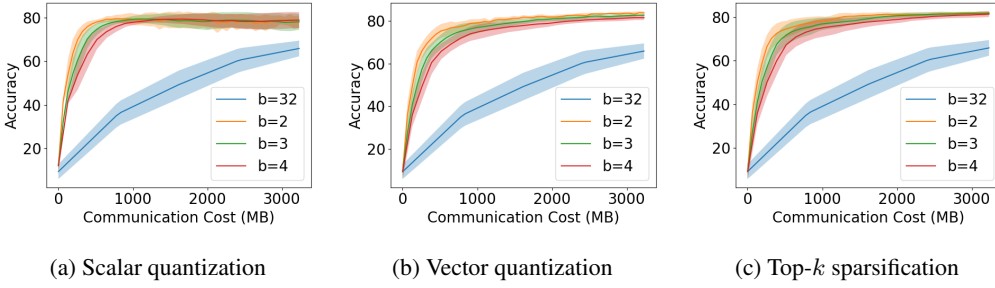

(a) Scalar quantization        (b) Vector quantization        (c) Top-$k$ sparsification

Figure A.10: Test accuracy on ModelNet10 dataset plotted by communication cost. We can see that all compressors reach higher accuracies with lower communication cost than no compression. We can see that when less bits are used in each compressor, higher test accuracies are reached at lower communication costs. Scalar quantization generally performs worse than the other compressors on the ModelNet10 dataset.

Figures A.9 and A.10 plot the test accuracy for training on the ModelNet10 dataset against the communication cost. The plots in Figure A.9 include all compression techniques for a given level of compression, while the plots in Figure A.10 include all levels of compression for a given compression technique. We can see that all compressors reach higher accuracies with lower communication cost than no compression. Scalar quantization generally performs worse than the other compressors on the ModelNet10 dataset. From Figure A.10, we also see that when fewer bits are used in each compressor, higher test accuracies are reached at lower communication costs.

