# OpenReview forum: "Compressed-VFL: Communication-Efficient Learning with Vertically Partitioned Data"
_ICLR.cc/2022/Conference — ICLR 2022 Submitted_

### Official Review · Reviewer_2WsW · 2021-11-02

**Correctness:** 4
**Technical Novelty And Significance:** 2
**Empirical Novelty And Significance:** 2
**Recommendation:** 6
**Confidence:** 3

**Main Review:**

## PAPER SUMMARY

This paper presents a new vertical federated learning algorithm that supports compression, in order to save communication cost in practice.
In particular, this paper proves that the proposed algorithm converges with a similar rate as the original algorithm without compression.
In vertical federated learning, each party involved does not have all the features, but only a subset of them.

The compression technique can be various, including top-k sparsification and quantization.

The compression error has to be diminishing along the training process.

## NOVELTY & SIGNIFICANCE

Applying compression on activation of some layers of a neural network while keeping the convergence rate has been studied in different scenarios for a while.  This includes model parallelism (the split learning referenced in this paper and the vertical federated learning problems are both special case of general model parallelism).

How such compression techniques work in the context of vertical federated learning is still not well studied yet. I appreciate the authors' work to actually show the convergence and experimental results.

## TECHNICAL SOUNDNESS

Since the compression error goes to zero along the training process, the error coming from stochastic gradients dominates and the convergence should be similar to ordinary stochastic gradient algorithm asymptotically.

Assumption 5 assumes a bounded gradient, which seems to be a bit strong to me (many existing compressed stochastic algorithms do not need such bounded gradient assumptions).

## OTHERS

Experiments look fine. The reference list format needs to be fixed (not consistent now).


**Summary Of The Paper:**

compression applied on vertical federated learning problems

**Summary Of The Review:**

An incremental work, by applying compression on vertical federated learning algorithm. The theoretical analysis follows existing compression stochastic algorithms' analysis (where the compression error eventually goes to zero).

---

> ### Author Response · Authors · 2021-11-23
> **Response to Reviewer 2WsW**
>
> Thank you for taking the time to review our paper.
> We appreciate that you see the value in our contributions.
> We address your concerns below:
>
> > Assumption 5 assumes a bounded gradient, which seems to be a bit strong to me (many existing compressed stochastic algorithms do not need such bounded gradient assumptions).
>
> Previous work on compression in Federated Learning
> focuses on the HFL setting, where gradients are compressed. There is no existing work that studies the convergence of VFL algorithms that use embedding compression. Embeddings are parameters in the partial derivatives calculated at each party. The effect of compressing embeddings on the partial derivatives in VFL and the effect of compressing gradients in HFL are different; therefore, the analysis in prior work on gradient compression in HFL does not apply to compression in VFL.
>
> To further clarify, Assumption 5 is not the standard bounded gradient assumption on the loss function. The assumption only bounds the embedding gradients. We do not require the loss function gradient to be bounded.
> When embeddings are compressed, this introduces error that propagates during the calculations of the partial derivatives of the loss function at each party. In order to bound this error, the embedding gradient must be bounded. Assumption 5 is necessary, and is not just for simplicity of analysis. We have revised the paper to note this in Section 2 under the assumptions.
>
> We summarize how our convergence analysis depends on Assumptions 4 and 5 as follows.
> We bound the compression error by use of the chain rule and Taylor series expansion. Assumption 4 helps us bound the higher-order terms in the Taylor series, while Assumption 5 helps bound the embedding gradient that appears as a coefficient due to the chain rule.
>
> This can be seen in the proof of Lemma 1 and Lemma 1'.
> Looking at the proof of Lemma 1, in (A.21) we apply the chain rule, and take the Taylor series expansion in (A.22) in order to separate the compression error from the gradient computation. We then can use Lemma 1' to bound this additive error.
> In (A.17), we note that the terms in $R_0^m()$ can be bounded using Assumption 4, but the coefficient in (A.16) depends on the embedding gradient, and must be upper bounded in order to ensure convergence.
>
>
> > The reference list format needs to be fixed (not consistent now).
>
> Thank you for pointing this out. We have updated the references section to make the formatting consistent throughout.

---

### Official Review · Reviewer_aP4B · 2021-11-04

**Correctness:** 2
**Technical Novelty And Significance:** 2
**Empirical Novelty And Significance:** Not applicable
**Recommendation:** 5
**Confidence:** 5

**Main Review:**

Strengths:
Communication is one of the major bottlenecks in Federated Learning, though the problem is less severe for the VFL compared to HFL for it only transfers embeddings. It may need a communication efficient algorithm as the embeddings may be in very high dimension. The paper also provides theoretical analysis of the algorithm and proves the algorithm can achieve the normal SGD-style convergence rate. The experiments also show that the algorithm can reduce the communication cost compared to the vanilla VFL algorithm.

Weakness:
There are some questions to be answered:
1.	What’s your assumption about label ownership, it seems that every client requires access to label y? This is not realistic in practice. The VFL typically avoids such situation.
2.	You mentioned that unbiasedness is maintained even if the client uses the same batch of samples during the whole local training, this is not self-evident. In eq.(7), I agree the equality holds for the step $t_0 +1$, what about $t > t_0 + 1$? Since the previous state $\Phi_m^{t-1}$ relies on the batch $B$, how can we get the equality?
3.	The experimental results (Fig.2) show that the proposed algorithm reduces total communication, however this is not obvious theoretically. As shown in the analysis, the compression error needs to approach 0 as $T$ increase, which means that the transferred dimension aproaches $P_m$. So can you provide some analysis of the total communication cost? does it save a constant factor of communication or with a larger factor?



**Summary Of The Paper:**

This paper proposes a communication-efficient training algorithm for vertical federated learning (VFL). It compresses the embeddings (clients) and parameters (the server) to save communication, what’s more, it also performs multiple local epochs to save more communication. Then it proves convergence rate of O(1/{T}^0.5) for the proposed algorithm. Finally, it provides some empirical results for validation.

**Summary Of The Review:**

The paper studies an interesting problem. The proposed algorithm attains theoretical guarantees, and its performance is also empirically validated. However, there are still some questions requiring more discussion.

---

> ### Author Response · Authors · 2021-11-23
> **Response to Reviewer aP4B**
>
> Thank you for taking the time to review our paper. We have addressed each of your concerns below:
>
> > What’s your assumption about label ownership, it seems that every client requires access to label y? This is not realistic in practice. The VFL typically avoids such situation.
>
>
> As discussed in the 2nd paragraph of Section 2, we assume each party has a copy of the labels $y$. In low-risk scenarios, like predicting credit score, labels may not need to be private among the parties.
>
> We agree this is not always the case in VFL, and have submitted a revision with a new paragraph in the Privacy subsection of Section 3 to discuss label privacy.
> For cases where $y$ is private and available only at a single party,
> For some classes of model architectures, Liu et al. [2019] provide a method for gradient calculation without the need for sharing labels.
> This method can be run within C-VFL by adding additional communication with the label holder at the start of each global round. Our analysis would still hold in this case, though the communication cost would increase. This communication cost, however, is reduced by the use of message compression.
>
> > You mentioned that unbiasedness is maintained even if the client uses the same batch of samples during the whole local training, this is not self-evident. In eq.(7), I agree the equality holds for the step $t_0+1$, what about $t>t_0$? Since the previous state $\Phi_m^{t-1}$ relies on the batch $B$, how can we get the equality?
>
> There was indeed a mistake in (7), as according to Assumption 2 the stochastic gradient is only conditionally unbiased at time $t_0$:
>
> $$\mathbb{E}_{B^{t_0}}[\nabla_m F_B (\Phi_m^{t_0}) | \Theta^{\tau}, \tau=0,\ldots,t_0] = \nabla_m F( \Phi_m^{t_0} ).~~~(*)$$
>
> We have submitted a revision where we have updated our analysis to account for the biased nature of the gradients after time $t_0$. We have introduced a new lemma (Lemma 2 in the appendix of the revised version) that bounds the difference between the gradients at the start of each global round and those calculated during local iterations on the reused mini-batch.
> The proof of this lemma does rely on a slightly revised version of Assumption 1. This assumption is common in SGD convergence analyses [Nguyen et al. 2018, Gower et al., 2019].
>
> We apply $(*)$ to the gradients at the start of each global round
> and then use Lemma 2 to upper bound the gradients in each local iteration that reuses a mini-batch.
> As a result, some of the error terms in Theorem 1 have changed, but the asymptotic result and the main takeaways remain the same.
>
> > The experimental results (Figure 2) show that the proposed algorithm reduces total communication, however this is not obvious theoretically. As shown in the analysis, the compression error needs to approach 0 as $T$ increase, which means that the transferred dimension aproaches $P_m$. So can you provide some analysis of the total communication cost? does it save a constant factor of communication or with a larger factor?
>
>
> Since the relationship between the communication cost and compression factor is different for each compressor, it is not simple to say how much communication is saved in general. However, in Remark 1, we point out that according to Theorem 1, C-VFL can afford some amount of compression error without worsening the asymptotic convergence rate. We then discuss this further in the Common Compressors section and Table 1 for specific compressors.
>
> In the first row of Table 1 we show how to choose common compressor parameters to achieve a convergence rate of $O(1/\sqrt{T})$. The values of $q$, $V$, and $k$ determine the compression ratio and thus, the overall communication savings. For known bounds on the embedding values, and a set number of iterations $T$, one can use this table to determine how much communication it is possible to save without slowing asymptotic convergence.
>
> In our revised submission, we have updated Remark 1, Table 1, and the common compressors section to clarify our analysis of the communication cost versus convergence rate.
>
> References
>
> Yang Liu et al., ``A communication efficient vertical federated learning framework." NeurIPS Workshop on Federated Learning for Data Privacy and Confidentiality, 2019.

---

### Official Review · Reviewer_JWDi · 2021-11-07

**Correctness:** 3
**Technical Novelty And Significance:** 2
**Empirical Novelty And Significance:** 2
**Recommendation:** 5
**Confidence:** 3

**Main Review:**

The main contribution of this paper is to offer the theoretical analysis of the effect message compression has on distributed training over vertically partitioned data, and prove convergence of non-convex objectives. This result is from a recent line of research that has been studied for a while. For example, Richtarik & Takac (2016); Hardy et al. (2017) were the first works to propose Federated Learning algorithms for vertically partitioned data. This work extended it to consider multiple local iterations, whose contribution is considered incremental.

The numerical experiment considered is also not enough, for example, the setup consists of four parties and a server. Even if we have real world examples of collaboration between a few institutions, such an experiment is still considered limited. The dataset used in the paper is also limited and small.




**Summary Of The Paper:**

This work studies the problem of communication-efficient training on vertically partitioned data and it is relevant to the conference. The paper provides the theoretical analysis of the effect message compression has on distributed training over vertically partitioned data, and prove convergence of non-convex objectives to a fixed point at a rate of O($1/\sqrt{T}$) when the compression error is bounded over the course of training. Experimental validation is provided to show compression can reduce communication by over 90% without a significant decrease in accuracy over VFL without compression.


**Summary Of The Review:**

In short, the paper is technically sound and the developments are clear. The derived analysis under non-convex objectives seems to be a useful contribution to the literature, showing a modest improvement over the state of the art. ​However, the paper is still not novel enough based on several existing works and could be strengthened by demonstrating more significant results instead of incremental, such as weaker conditions.

---

> ### Author Response · Authors · 2021-11-23
> **Response to Reviewer JWDi (1/2)**
>
> Thank you for taking the time to review our paper. We have addressed each of your concerns below:
>
> > The main contribution of this paper is to offer the theoretical analysis of the effect message compression has on distributed training over vertically partitioned data, and prove convergence of non-convex objectives. This result is from a recent line of research that has been studied for a while. For example, Richtarik and Takac (2016); Hardy et al. (2017) were the first works to propose Federated Learning algorithms for vertically partitioned data. This work extended it to consider multiple local iterations, whose contribution is considered incremental.
>
>
> We appreciate the acknowledgment of our theoretical contribution.
>
> We agree that the consideration of multiple local iterations and a non-linear server model are natural extensions to the VFL framework. The effect of multiple local iterations in VFL with a linear server model was studied by [Liu et al., 2019] previously, and the effects of a non-linear server model in VFL without multiple local iterations was studied by [Chen et al., 2020].
> However, proving convergence with the combination of multiple local iterations, a non-linear server model, and embedding compression is more difficult than proving each individually.
>
> In particular, our analysis has the following novel aspects:
>
> - We model the compression error as additive perturbation to the embeddings and the server model.
>
> - We analyze the impact of these perturbations on the loss function, which is non-linear.
>
> - We analyze the impact of stale compressed embeddings in the context of VFL with multiple local iterations.
>
> By including all three of these features in C-VFL and proving convergence, we have extended the general VFL framework while also providing methods of analyzing convergence in many different variations of VFL and Split Learning.

---

> > ### Author Response · Authors · 2021-11-23
> > **Response to Reviewer JWDi (2/2)**
> >
> > > The numerical experiment considered is also not enough, for example, the setup consists of four parties and a server. Even if we have real world examples of collaboration between a few institutions, such an experiment is still considered limited. The dataset used in the paper is also limited and small.
> >
> >
> > In the original submission, we included an experiment with $12$ parties and a server to illustrate a case with more parties (see Figure 3b). We agree it would be good to see our algorithm tested on more parties and larger datasets. Thus, in our revised submission, we have included two new sets of experiments, the first with $48$ parties, and the second with the CIFAR-10 dataset. The results from these experiments can be found in the appendix in Section A.4.
> >
> > In the first experiment, C-VFL is run with $48$ parties training with the ModelNet10 dataset. Every $4$ parties are assigned a view, and the parties each store a quadrant of the image. In Figure A.1, we plot the test accuracy for the ModelNet10 dataset. The algorithm converges but the test accuracy is overall lower than when running with $4$ and $12$ parties as predicted by Theorem 1. This is expected, as each party has less information
> > individually, making the prediction task more difficult. Figure A.1a shows the test accuracy plotted by epochs. There is very little variation between compressors and no compression here. There is a very large benefit for reducing communication cost when using compression, seen in Figure A.1b. In Figure A.1c, we plot the test accuracy of C-VFL using vector quantization for different values of $b$, the number of bits to represent compressed values. Similar to previous results, lower $b$ tends to improve test accuracy reached with the same amount of communication cost.
> >
> > For the second experiment, to simulate a VFL scenario with the CIFAR-10 dataset, we split the images into 4 quadrants and ran C-VFL with 4 parties and a server. Each party trains ResNet-18 locally, and the server model is a single fully-connected layer.
> > In Figure A.2, we plot the test accuracy for the CIFAR-10 dataset.
> > Again, C-VFL converges but the test accuracy is fairly low compared to typical baseline accuracies, which is expected, as object classification is difficult with vertical partitioning. Figure A.2a shows the test accuracy plotted by epochs. We can see that vector quantization performs almost as well as no compression in the CIFAR-10 dataset. When plotting by communication cost, seen in Figure A.2b, we can see that vector quantization performs the best, though scalar quantization and top-k sparsification show communication savings as well. In Figure A.2c, we plot the test accuracy of C-VFL using vector quantization for different values of $b$, the number of bits to represent compressed values. Similar to previous results, lower $b$ tends to improve test accuracy reached with the same amount of communication cost.
> >
> > We include these experiments in the appendix rather than in Section 5 as the ModelNet10 and MIMIC-III datasets are common in VFL benchmarking [Liu et al. 2019, Chen et al. 2020], and these datasets are typically partitioned among $4$ or $12$ parties.
> >
> > References
> >
> > Yang Liu et al., ``A communication efficient vertical federated learning framework." NeurIPS Workshop on Federated Learning for Data Privacy and Confidentiality, 2019.
> >
> > Chen et al., ``VAFL: a method of vertical asynchronous federated learning." arXiv, 2020.

---

### Official Review · Reviewer_JhoT · 2021-11-07

**Correctness:** 2
**Technical Novelty And Significance:** 3
**Empirical Novelty And Significance:** 2
**Recommendation:** 5
**Confidence:** 3

**Main Review:**

This paper studies an interesting and important problem. Frequent communication across different parties is expensive and it is natural to consider using local iterations. But compressing embeddings is not well studied and it was not clear how it affect the convergence. Overall, the paper is well written but it lacks of more detailed description of the compression methods it considers. Table 1 gives the error bound for scalar quantizer, vector quantizer, and top-k sparsification but the their algorithmic details are omitted. Though I think the paper is interesting, but I have some questions regarding theory:

1. The update of $\theta_m^t$ depends on the mini-batch $B$. How can we say that the gradient is conditionally unbiased for any $t > t_0$? Especially, the RHS of (7) still depends on the realization of $B$. This a critical prerequisite for establishing the convergence in the paper.
2. The bound (8) in Lemma 1 increases linearly with the mini-batch size, which is loose.
3. How do we select the compressor so that compression error decreases at the same rate as the learning rate? This is crucial to ensure the convergence but there is no discussion.
4. Error bound (9) has a quadratic dependency on the number of parties $M$, which also seems loose to me.
5. It is good to provide a reference convergence bound for the case where the compression is not adopted.

**Summary Of The Paper:**

There is a growing interest in vertical federated learning where each party only store a subset of features due to its various important applications. This paper studies how to make vertical federated learning more efficient. It proposed to apply compression to the embeddings shared periodically. Compression has been well studied for reducing the network traffic for synchronizing gradient but there is little study in compressing embeddings. This paper shows that the convergence is guaranteed if the compression errors diminish at the same rate as the learning rate. The experimental results show that the vector quantizer produces the same accuracy as the full-precision counterpart while significantly reducing the communication costs.

**Summary Of The Review:**

This paper studied an important problem, but the theoretical results are not convincing.

---

> ### Author Response · Authors · 2021-11-23
> **Response to Reviewer JhoT (1/2)**
>
> Thank you for taking the time to review our paper. We have addressed each of your concerns below:
>
> > The update of $\theta_m^t$ depends on the mini-batch $B$. How can we say that the gradient is conditionally unbiased for any $t>t_0$? Especially, the RHS of (7) still depends on the realization of $B$. This a critical prerequisite for establishing the convergence in the paper.
>
> There was indeed a mistake in (7), as according to Assumption 2 the stochastic gradient is only conditionally unbiased at time $t_0$:
>
> $$\mathbb{E}_{B^{t_0}}[\nabla_m F_B (\Phi_m^{t_0}) | \Theta^{\tau}, \tau=0,\ldots,t_0] = \nabla_m F( \Phi_m^{t_0} ).~~~(*)$$
>
> We have submitted a revision where we have updated our analysis to account for the biased nature of the gradients after time $t_0$. We have introduced a new lemma (Lemma 2 in the appendix of the revised version) that bounds the difference between the gradients at the start of each global round and those calculated during local iterations on the reused mini-batch.
> The proof of this lemma does rely on a slightly revised version of Assumption 1. This assumption is common in SGD convergence analyses [Nguyen et al. 2018, Gower et al., 2019].
>
> We apply $(*)$ to the gradients at the start of each global round
> and then use Lemma 2 to upper bound the gradients in each local iteration that reuses a mini-batch.
> As a result, some of the error terms in Theorem 1 have changed, but the asymptotic result and the main takeaways remain the same.
>
> > The bound (8) in Lemma 1 increases linearly with the mini-batch size, which is loose.
>
> Thank you for pointing this out. This was a by-product of our definition of $e$ being the maximum over the mini-batch and simplifying instances of $\epsilon$ with the upper bound $e$. In the revised paper, we have defined the variable
> $\mathcal{E}_m^t = \mathbb{E} || \epsilon_m^t ||^2$ which captures the total expected squared compression error from a party $m$ at time $t$. This change allows us to show the exact compression error rather than using a loose upper bound.
>
> In the revised paper, we have rewritten Theorems 1 and 2, Remark 1, and our analysis in appendix section A.1 with $\mathcal{E}_m^t$ rather than simplifying using $e_m^t$ as a bound.
> The bound in Lemma $1$ no longer increases linearly with the mini-batch size.
> We can see in the second row of Table 1 that the compression error may still grow linearly with the mini-batch size $B$, depending on the compressor used in C-VFL. However, for such compressors, we can choose compressor parameters as shown in the first row of Table 1 to still ensure a convergence rate of $O(\frac{1}{\sqrt{T}})$.
>
> > How do we select the compressor so that compression error decreases at the same rate as the learning rate? This is crucial to ensure the convergence but there is no discussion.
>
> In the revision, we have updated the discussion that follows Theorem 2 to clarify the relationship between the step size and compression error.
> Further, we have updated the Common Compressors subsection to explain how one can choose compressor parameters to ensure convergence.
>
> In essence, according to Theorem 2, the product of the step size and the compression error must be summable over all iterations.
> Let $\eta^{t_0} = O(1/t_0)$, where $t_0$ is the current global round. Then setting $T = t_0$
> in Table 1 provides a choice of compression parameters at each iteration
> to ensure the compression error diminishes at a rate of $O(1/\sqrt{t_0})$, guaranteeing convergence.

---

> > ### Author Response · Authors · 2021-11-23
> > **Response to Reviewer JhoT (2/2)**
> >
> > > Error bound (9) has a quadratic dependency on the number of parties $M$, which also seems loose to me.
> >
> > In our revision, we have improved the bound with respect to $M$ in Theorem 1. The stochastic gradient variance no longer has a quadratic dependency on the number of parties $M$. We do still see a quadratic dependency on $M$ in the third term with the compression error $\mathcal{E}_j^t$.
> > The quadratic dependency of the compression error on $M$ arises from the difference between the global model and each party's compressed view of the global model. Specifically, all $M$ parties and the server each have a compressed view of the $M$ other parties.
> > For more details on this quadratic dependency, see the additional notation section in the appendix where we define $E_m^t$, and (A.91) where we apply the definition in our proof.
> >
> > Considering that $M$ is often small in the VFL setting [Kairouz et al. 2021], the quadratic dependency on $M$ is a minor contributor towards the overall convergence error. We have added a comment on this in the revision in Section 4 after Theorem 1.
> >
> >
> > > It is good to provide a reference convergence bound for the case where the compression is not adopted.
> >
> > Thank you for the suggestion!
> > Liu et al. [2019] provide a bound for a similar VFL algorithm without compression. The convergence rate in our Theorem 1 is an improvement over this bound. Specifically, in [Liu et al., 2019], the batch size needs to increase with $T$ to guarantee convergence, while we do not have this requirement.
> > Further, their bound has a quadratic dependency on $M$ and $Q$ while our bound has a linear dependency on each when the compression error is zero.
> > We have updated the discussion after Theorem 1 to clarify this.
> >
> >
> > References
> >
> > Yang Liu et al., ``A communication efficient vertical federated learning framework." NeurIPS Workshop on Federated Learning for Data Privacy and Confidentiality, 2019.
> >
> > Kairouz et al., ``Advances and open problems in federated learning." Foundations and Trends in Machine Learning, 2021.
> >
> > Nguyen et al., ``SGD and hogwild! convergence without the bounded gradients assumption." International Conference on Machine Learning, 2018.
> >
> > Gower et al., ``SGD: General Analysis and Improved Rates" International Conference of Machine Learning, 2019.

---

### Author Response · Authors · 2021-11-29
**Responses to Reviewers and Paper Revision**

We thank all the reviewers again for taking time to review our paper. We have responded to each of your concerns in the comments below and revised our paper accordingly. As the discussion period ends today, could you kindly let us know whether there are further questions after reading our responses and revision? If not, would you kindly consider increasing your review scores? Thank you!

---

### Decision · Program_Chairs · 2022-01-20

**Decision:**

Reject

**Comment:**

Reviewers have all agreed that this paper studied an important problem and made valuable contributions. The goal is to reduce the communication costs  of Federated learning where the data are stored in different parities based on subsets of features.  The paper developed the theory to show guaranteed convergence and provided empirical evaluations to validate the theory.

On the other hand, compared with existing literature, Reviewers feel that the novelty of this submission appears limited and the improvements seem to be incremental. Reviewers appreciate the Authors' efforts in conducting the detailed rebuttals and providing an improved manuscript. We hope the authors would continue to improve the paper based on reviews, when they prepare for their future submission.